# Plateau in Monotonic Linear Interpolation — A "Biased" View of Loss Landscape for Deep Networks

**Xiang Wang[†], Annie N. Wang[§], Mo Zhou[†], Rong Ge[†]**
Department of Computer Science
Duke University, USA
[†]{xwang,mozhou,rongge}@cs.duke.edu,[§]annie.wang029@duke.edu

## Abstract

Monotonic linear interpolation (MLI) — on the line connecting a random initialization with the minimizer it converges to, the loss and accuracy are monotonic — is a phenomenon that is commonly observed in the training of neural networks. Such a phenomenon may seem to suggest that optimization of neural networks is easy. In this paper, we show that the MLI property is not necessarily related to the hardness of optimization problems, and empirical observations on MLI for deep neural networks depend heavily on the biases. In particular, we show that interpolating both weights and biases linearly leads to very different influences on the final output, and when different classes have different last-layer biases on a deep network, there will be a long plateau in both the loss and accuracy interpolation (which existing theory of MLI cannot explain). We also show how the last-layer biases for different classes can be different even on a perfectly balanced dataset using a simple model. Empirically we demonstrate that similar intuitions hold on practical networks and realistic datasets.

## 1 Introduction

Deep neural networks can often be optimized using simple gradient-based methods, despite the objectives being highly nonconvex. Intuitively, this suggests that the loss landscape must have nice properties that allow efficient optimization. To understand the properties of loss landscape, Goodfellow et al. (2014) studied the linear interpolation between a random initialization and the local minimum found after training. They observed that the loss interpolation curve is monotonic and approximately convex (see the MNIST curve in Figure 1) and concluded that these tasks are easy to optimize. However, other recent empirical observations, such as Frankle (2020) observed that for deep neural networks on more complicated datasets, both the loss and the error curves have a long plateau along the interpolation path, i.e., the loss and error remain high until close to the optimum (see the CIFAR-10 curve in Figure 1). Does the long plateau along the linear interpolation suggest these tasks are harder to optimize? Not necessarily, since the hardness of optimization problems does not need to be related to the shape of interpolation curves (see examples in Appendix A).

In this paper we give the first theory that explains the plateau in both loss and error interpolations. We attribute the plateau to simple reasons as the bias terms, the network initialization scale and the network depth, which may not necessarily be related to the difficulty of optimization.

Note that there are many different theories for the optimization of overparametrized neural networks, in particular the neural tangent kernel (NTK) analysis (Jacot et al., 2018; Du et al., 2018; Allen-Zhu et al., 2019; Arora et al., 2019) and mean-field analysis (Chizat & Bach, 2018; Mei et al., 2018). However they don't explain the plateau in both loss and error interpolations. For NTK regime, the network output is nearly linear in the parameters and the loss interpolation curve is monotonically decreasing and convex — no plateau in the loss interpolation. Mean-field regime often uses a smaller initialization on a homogeneous neural network (as considered in Chizat & Bach (2018); Mei et al. (2018)). In this case, the interpolated network output is basically a scaled version of the network output at the minimum and has same label predictions — no plateau in the error interpolation curve.

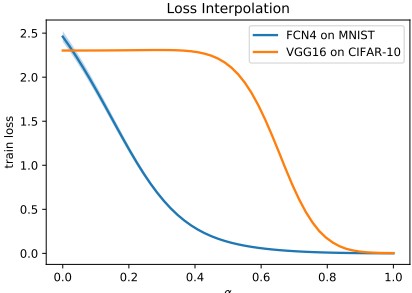 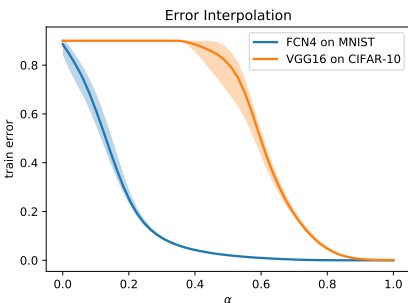

Figure 1: Loss interpolation curve and error interpolation curve for a four-layer fully-connected network (FCN4) on MNIST and for VGG16 on CIFAR-10.

## 1.1 OUR RESULTS

Our theoretical results consist of two parts. In the first part (see Section 3), we give a plausible explanation for the plateau in both the loss and error curves.

**Claim 1** (informal). *If a deep network has a relatively small initialization, and its last-layer biases are significantly different for different classes, then both the loss and error curves will have a plateau. The length of the plateau is longer for a deeper network.*

We formalize this claim in Theorem 1. For intuition, consider an $r$-layer neural network that only has bias on the last layer, and consider Xavier initialization (Glorot & Bengio, 2010) which typically gives small output and zero bias. If we consider the $\alpha$-interpolation point (with coefficient $\alpha$ for the minimum and $(1 - \alpha)$ for the initialization), then the weight "signal" from the minimum scales as $\alpha^r$ (as it is the product of $r$ layers) while the bias scales as $\alpha$. As illustrated in Figure 2 (right), when $r$ is large and there is a difference in biases, the bias will dominate, which creates a plateau in error. For the loss, one can also show that the weight signal is near 0 for small $\alpha$, so the network output is dominated by the biases and the loss cannot beat the random guessing at initialization. Note that this explanation for the plateau does not have any implication on the hardness of optimization.

However, why would the last-layer biases be different for different classes, especially in cases when the biases are initialized as zeros and all classes are balanced? In the second part (see Section 4), we focus on a simple model that we call $r$-homogeneous-weight network. This is a two-layer network whose $i$-th output is $\langle W_{i,:}, x \rangle^r + b_i$, where $x \in \mathbb{R}^d$ is the network input, $W_{i,:} \in \mathbb{R}^d$ is the weight vector and $b_i \in \mathbb{R}$ is the bias (see Figure 2 (left)). Our simple model simulates a depth-$r$ ReLU/linear network with bias on the output layer, in the sense that the signal is $r$-homogeneous while the bias is 1-homogeneous in the parameters. Under this model we can show that:

**Claim 2** (informal). *For the $r$-homogeneous-weight network on a simple balanced dataset, the class that is learned last has the largest bias.*

Here, a class is learned when all the samples in this class get classified correctly with good confidence. We basically show that once a class gets learned, the bias associated with this class starts decreasing and eventually the class that is learned last has the largest bias. We formalize this claim in Theorem 2.

In Section 5, we verify these ideas empirically on fully-connected networks for MNIST (Deng, 2012), Fashion-MNIST (Xiao et al., 2017) and on VGG-16 (Simonyan & Zisserman, 2014) for CIFAR-10, CIFAR-100 (Krizhevsky et al., 2009). We first show that if we train a neural network *without* using any bias, then the error curve has much shorter plateau or no plateau at all. Even for networks that are trained normally with biases, we design a homogeneous interpolation scheme for biases to make sure that both biases and weights are $r$-homogeneous. Such an interpolation indeed significantly shortens the plateau for the error. We also show that decreasing the initialization scale or increasing the network depth can produce a longer plateau in both the error and loss curves. Finally, we show that the bias is correlated with the ordering in which the classes are being learned for small datasets, which suggests that even though the model we consider in the convergence analysis is simple, it captures some of the behavior in practice.

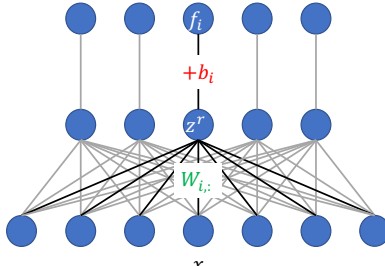 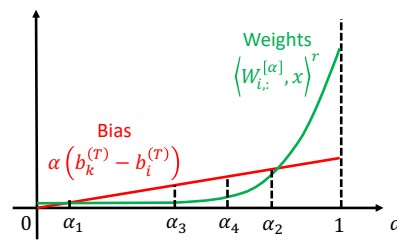

Figure 2: **(Left)** Our $r$-homogeneous-weight model with $f_i(x) = \langle W_{i,:}, x \rangle^r + b_i$. **(Right)** The comparison between interpolated bias $\alpha(b_k^{(T)} - b_i^{(T)})$ and interpolated weights signal $\left\langle W_{i,:}^{[\alpha]}, x \right\rangle^r$.

### 1.2 RELATED WORKS

There are two major lines of work studying interpolation between different points for neural networks, one on monotonic linear interpolation that interpolates the initial network and the learned network, and the other on mode connectivity that connects two learned networks.

**Monotonic linear interpolation.** Goodfellow et al. (2014) first studied the linear interpolation between the network at initialization and the network after training on MNIST. Frankle (2020) extended the experiments to modern networks on CIFAR-10 and ImageNet and found that though the loss/error is still monotonically non-increasing along the path, it remains high until close to the optimum. Lucas et al. (2021) showed that MLI holds when the network output curve along the interpolation path is close to linear (measured by Gaussian length). However, the Gaussian length can only be formally controlled in the NTK regime.

**Mode connectivity.** Mode connectivity considers the interpolation between two learned networks (modes) found by SGD. In general, a linear interpolation between two different local minima crosses regions of high loss (Goodfellow et al., 2014). Surprisingly, Draxler et al. (2018) and Garipov et al. (2018) observed that local minima found by SGD from different initializations can be connected via a piece-wise linear path of low loss. Frankle et al. (2020) and Fort et al. (2020) observed that local minima trained from the same initialization can also be connected using a linear path. Freeman & Bruna (2016); Venturi et al. (2018); Nguyen (2019; 2021); Kuditipudi et al. (2019); Shevchenko & Mondelli (2020); Nguyen et al. (2021) gave several theoretical explanations for this phenomenon.

### 2 PRELIMINARIES

We first formally define the linear interpolation between the network at initialization and the network after training. Then we describe the notations that we will use in the paper.

**Linear interpolation:** Consider a network with parameters $\theta \in \mathbb{R}^p$. Suppose the network is initialized with parameters $\theta^{(0)}$ and it converges to $\theta^{(T)}$. A *linear interpolation* is constructed by setting the parameters $\theta^{[\alpha]} = (1 - \alpha)\theta^{(0)} + \alpha\theta^{(T)}$ for $\alpha \in [0, 1]$. The *loss interpolation curve* is defined as $\gamma_{\text{loss}}(\alpha) : [0, 1] \to \mathbb{R}$ such that $\gamma_{\text{loss}}(\alpha)$ is the training loss of the network at $\theta^{[\alpha]}$. Similarly, the *error interpolation curve* is defined as $\gamma_{\text{error}}(\alpha) : [0, 1] \to [0, 1]$ with $\gamma_{\text{error}}(\alpha)$ as the training error of the network at $\theta^{[\alpha]}$. Here, the training error is simply the ratio of training samples that get classified incorrectly by the network.

**Notations:** We use $[k]$ to denote the set $\{1, 2, \cdots, k\}$. We use $\mathcal{N}(0, \delta^2)$ to denote the Gaussian distribution with mean zero and variance $\delta^2$. We use $\|\cdot\|$ to denote the $\ell_2$ norm for a vector or the spectral norm for a matrix. For any non-zero vector $v$, we use $\bar{v}$ to denote $v/\|v\|$. We use $O(\cdot), \Theta(\cdot), \Omega(\cdot)$ to hide the dependency on constant factors and use $\widetilde{O}(\cdot), \widetilde{\Theta}(\cdot), \widetilde{\Omega}(\cdot)$ to hide the dependency on poly-logarithmic factors.

For any time $t$, we use $\theta^{(t)}, f^{(t)}$ to denote the parameters and the network at time $t$. For any $\alpha \in [0, 1]$, we use $\theta^{[\alpha]}, f^{[\alpha]}$ to denote the $\alpha$ interpolation point, which means $\theta^{[\alpha]} := (1 - \alpha)\theta^{[0]} + \alpha\theta^{[T]}$ and $f^{[\alpha]}$ is the network with parameters $\theta^{[\alpha]}$.

## 3 PLATEAU FOR LOSS AND ERROR INTERPOLATIONS

We prove that the long plateau exists in the loss and error curves when the initialization is small and the network is deep on fully-connected networks. The detailed proof can be found in Appendix B.3.

We consider an $r$-layer fully-connected neural network with $r$ at least three. Given input $x \in \mathbb{R}^{n_0}$, the network output is

$$g(x) := V_r \sigma \left( V_{r-1} \cdots \sigma(V_1 x) \cdots \right) + b, \tag{1}$$

where $V_i \in \mathbb{R}^{n_r \times n_{r-1}}$ for each layer $i \in [r]$ and $b \in \mathbb{R}^{n_r}$. Here the activation function $\sigma(\cdot)$ can be either identity function or ReLU function. The output layer width equals to the number of classes, i.e., $n_r = k$. We use $L(\{V_i\}, b)$ to denote the sum of cross entropy loss over all samples.

For the biases, we initialize them as zeros and assume after training there exists a gap between the largest bias and the second largest, which also holds empirically (see Figure 8). Note this bias gap is essential for the plateau in the error interpolation. If all the biases are equal in the trained network, the logits for different classes only differ by the weights signal and the interpolated network has same label predictions as the trained network.

**Assumption 1** (Bias Gap). *Choosing $i^* \in \arg\max_{i \in [k]} b_i^{(T)}$, we have $b_{i^*}^{(T)} - \max_{i \neq i^*} b_i^{(T)} > 0$. Without loss of generality, we assume that $b_k^{(T)} > \max_{i \in [k-1]} b_i^{(T)}$. We denote $\Delta_{\min} := b_k^{(T)} - \max_{i \in [k-1]} b_i^{(T)}$ and $\Delta_{\max} := b_k^{(T)} - \min_{i \in [k-1]} b_i^{(T)}$.*

Then, we show both the loss and error interpolation curves have a long plateau in Theorem 1.

**Theorem 1.** *Suppose the network is defined as in Equation (1) and suppose the weights satisfy $\left\| V_i^{(0)} \right\| \leq \delta, \left\| V_i^{(T)} \right\| \leq V_{\max}$ for all layers $i \in [r]$. On a $k$-class balanced dataset whose inputs have $\ell_2$ norm at most 1, if Assumption 1 holds, for any $\epsilon > 0$, as long as $\delta < \min\left( \frac{\epsilon^{1/r}}{r}, \frac{1}{r^2}, \left( \frac{1}{2e} \right)^{\frac{2}{r-2}} \right)$, there exist $\alpha_1 = \frac{\delta}{\Delta_{\min}}, \alpha_2 = \left( \frac{1}{1+\sqrt{\delta}} \right)^{\frac{r}{r-1}} \left( \frac{\Delta_{\min}}{2V_{\max}^r} \right)^{\frac{1}{r-1}}$ and $\alpha_3 = \frac{\epsilon^{1/r}}{V_{\max}}$ such that*

1. *for all $\alpha \in [\alpha_1, \alpha_2]$, the error is $1 - 1/k$;*

2. *for all $\alpha \in [0, \alpha_3]$, we have $\log k - 2e\epsilon \leq \frac{1}{N} L\left( \left\{ V_i^{[\alpha]} \right\}, b^{[\alpha]} \right) \leq \log k + \alpha \Delta_{\max} + 2e\epsilon$, where $N$ is the number of training examples.*

The above theorem shows that for all $\alpha \in [\alpha_1, \alpha_2]$, the error remains at $1 - 1/k$ that is the same as random guessing. We skip the very short initial region $[0, \frac{\delta}{\Delta_{\min}}]$ since the bias is very small and the error can be unpredictable due to the randomness in initial weights. When initialization scale $\delta$ is small, this error plateau region is roughly $[0, (\frac{\Delta_{\min}}{2V_{\max}^r})^{\frac{1}{r-1}}]$. Empirically, $\frac{\Delta_{\min}}{2V_{\max}^r}$ is smaller than 1 and does not change much when depth increases. So the plateau becomes longer in a deeper network.

Intuitively, the plateau in error curve is there because for a small initialization, the output is close to $\alpha^r V_r^{(T)} \sigma\left( V_{r-1}^{(T)} \cdots \sigma(V_1^{(T)} x) \cdots \right) + \alpha b^{(T)}$. When $\alpha$ is not large enough $\alpha^r$ is much smaller than $\alpha$, so for every class $i \neq k$, the first term (signal part) cannot overcome the bias gap $\alpha(b_k^{(T)} - b_i^{(T)})$. This implies that all samples are predicted as class $k$ and the error is $1 - 1/k$.

We also show that the average loss cannot be lower than $\log k - 2e\epsilon$ when $\alpha \leq \frac{\epsilon^{1/r}}{V_{\max}}$. Note a small random initialization can achieve a loss of approximately $\log k$. Usually the bias gap $\Delta_{\max}$ in practice is not very large, so the loss curve remains nearly flat during this interpolation region. Again, the loss plateau is becoming longer when depth $r$ increases. This is because the weights signal remains near 0 for a larger range of $\alpha$.

## 4 TRAINING DYNAMICS FOR CREATING A BIAS GAP

In this section, we explain how the gradient flow dynamics generates a bias gap on a balanced dataset by analyzing a simple model. Below, we first define the network model, training dataset and optimization procedure for our analysis.

$r$**-homogeneous-weight network:** We consider a two-layer and $k$-output neural network with activation function $\sigma(z) := z^r$, where $r$ is a positive constant that is at least three. As illustrated in Figure 2 (left), under input $x \in \mathbb{R}^d$, the $i$-th output $f_i(x)$ is $\langle W_{i,:}, x \rangle^r + b_i$, where the weight vector $W_{i,:} \in \mathbb{R}^d$ is the $i$-th row of weight matrix $W \in \mathbb{R}^{k \times d}$ and $b_i \in \mathbb{R}$ is the $i$-th entry of vector $b \in \mathbb{R}^k$. In output $f_i(x)$, we call $\langle W_{i,:}, x \rangle^r$ the signal since it is input-dependent and call $b_i$ the bias.

**Dataset:** We consider a $k$-class balanced dataset, with $k$ as a constant. We denote the whole dataset as $\mathcal{S}$ and denote the subset for each class $i \in [k]$ as $\mathcal{S}_i$. Each subset $\mathcal{S}_i$ has exactly $N/k$ samples and each sample $x \in \mathbb{R}^d$ is independently sampled as $v_i + \xi$, where the noise $\xi \sim \mathcal{N}(0, \frac{\sigma^2}{d} I)$. To differentiate the noise terms among different samples, we denote the noise associated with sample $x$ as $\xi_x$. We assume all $v_i$'s are orthonormal; without loss of generality, we assume $v_i = e_i$ for each class $i$. Here, we assume the orthogonal features to facilitate the convergence analysis beyond the NTK regime, following previous works (Allen-Zhu & Li, 2020; Ge et al., 2021).

**Optimization:** We initialize each entry in weight matrix $W$ by independently sampling from Gaussian distribution $\mathcal{N}(0, \delta^2)$ and then taking the absolute value [1]. Our analysis can be trivially generalized to standard Gaussian initialization (without taking absolute value) when $r$ is an even integer. We initialize all bias terms as zeros. We use cross-entropy loss $L(W, b) = \sum_{i \in [k]} \sum_{x \in \mathcal{S}_i} - \log \left( \frac{\exp(f_i(x))}{\sum_{j \in [k]} \exp(f_j(x))} \right)$, and run gradient flow on $\frac{k}{N} L(W, b)$ for time $T$. Our analysis can also be extended to gradient descent with a small step size.

Next we show that running gradient flow from a small initialization can converge to a model with zero error and constant bias gap.

**Theorem 2.** *Suppose the neural network, dataset and optimization procedure are as defined in Section 4. Suppose initialization scale $\delta \leq \Theta(1)$, noise level $\sigma \leq \widetilde{\Theta}(1)$, dimension $d \geq \widetilde{\Theta}(1/\delta^{2r-2})$ and number of samples $N \geq \widetilde{\Theta}(1/\delta^{r-1})$, with probability at least $0.99$ in the initialization, there exists time $T = \Theta(\log(1/\delta)/\delta^{r-2})$ such that we have*

1. *zero error: for all different $i, j \in [k]$ and for all $x \in \mathcal{S}_i$, $f_i^{(T)}(x) \geq f_j^{(T)}(x) + \Omega(1)$;*

2. *bias gap: $b_{i*}^{(T)} - \max_{i \neq i^*} b_i^{(T)} \geq \Omega(1)$ with $i^* = \arg\max_{i \in [k]} b_i^{(T)}$.*

Due to space limit, we only give a proof sketch here and leave the detailed proof in Appendix C. Since our dataset is perfectly balanced, it might seem surprising that gradient flow learns diverse biases. We can compute the time derivative on the bias, $\dot{b}_i = 1 - \frac{k}{N} \sum_{x \in \mathcal{S}} u_i(x)$, where $u_i(x)$ is the softmax output for class $i$, that is $\frac{\exp(f_i(x))}{\sum_{i' \in [k]} \exp(f_{i'}(x))}$. At the beginning, all logits are small, we have $u_i(x) \approx 1/k$ and $\dot{b}_i \approx 0$. If all the samples are learned at the same time, we have $u_i(x) \approx 1, u_i(x') \approx 0$ for $x \in \mathcal{S}_i, x' \in \mathcal{S} \setminus \mathcal{S}_i$, which again leads to $\dot{b}_i \approx 0$.

On the other hand, we can consider what happens if all samples in one class (e.g., class $i$) are learned before any sample in any other class (e.g., class $j$) is learned [2]. In this case we have

$$\dot{b}_i = 1 - \frac{k}{N} \sum_{x \in \mathcal{S}_i} u_i(x) - \frac{k}{N} \sum_{x \in \mathcal{S} \setminus \mathcal{S}_i} u_i(x) \approx 1 - \frac{k}{N} \cdot \frac{N}{k} \cdot 1 - \frac{k}{N} \cdot \frac{N(k-1)}{k} \cdot \frac{1}{k} = -\frac{k-1}{k},$$

$$\dot{b}_j = 1 - \frac{k}{N} \sum_{x \in \mathcal{S}_i} u_j(x) - \frac{k}{N} \sum_{x \in \mathcal{S} \setminus \mathcal{S}_i} u_j(x) \approx 1 - \frac{k}{N} \cdot \frac{N}{k} \cdot 0 - \frac{k}{N} \cdot \frac{N(k-1)}{k} \cdot \frac{1}{k} = \frac{1}{k},$$

where for any learned sample $x \in \mathcal{S}_i$, we have $u_i(x) \approx 1, u_j(x) \approx 0$; for any not yet learned sample $x \in \mathcal{S} \setminus \mathcal{S}_i$, we have $u_i(x), u_j(x) \approx 1/k$. The above calculation shows that $b_i$ starts to decrease and all the other bias terms increase. Generalizing this intuition, we show that $b_{i'}$ starts to decrease whenever class $i'$ is learned, and the class that is learned last will have the largest bias.

---

[1] In the Xavier initialization, each entry in weight matrix $W$ is sampled from $\mathcal{N}(0, 1/d)$, so we can think of $\delta^2 = 1/d$ that is small when input dimension $d$ is large.

[2] This is indeed possible since all samples of one class only differ in the noise terms in our setting. In the analysis, we can show that the noise term has negligible contribution to the network output and all samples in one class are learned almost at the same time.

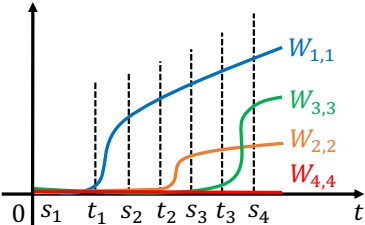 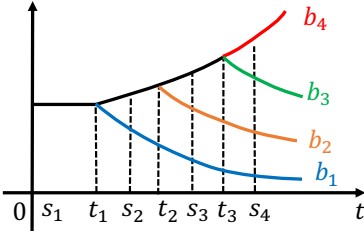

Figure 3: The training dynamics of $W$ and $b$ in a four-class example.

As the weights are initialized randomly, by standard anti-concentration, one can argue that there is a gap between $W_{i,i}^{(0)}$'s. Without loss of generality, we assume $W_{1,1}^{(0)} > W_{2,2}^{(0)} > \cdots > W_{k,k}^{(0)}$. The initial difference in the weights will lead to different classes being learned at different time. We show that by doing induction on the following hypothesis through training:

**Proposition 1** (Induction Hypothesis). *In the same setting of Theorem 2, with probability at least 0.99 in initialization, there exist time points $0 =: s_1 < t_1 < s_2 < t_2 < \cdots < s_{k-1} < t_{k-1} < s_k := T$ with $t_i - s_i = \Theta(\log(1/\delta)/\delta^{r-2})$ and $s_{i+1} - t_i = \Theta(1)$ for $i \in [k-1]$ such that for any $t \in [s_i, s_{i+1}]$,*

1. *(classes not yet learned) for any class $j, j' \geq i+1$, we have (1) $b_j^{(t)} \geq \max_{i' \in [k]} b_{i'}^{(t)} - O(\delta^r)$, (2) $\left| b_j^{(t)} - b_{j'}^{(t)} \right| \leq O(\delta^r)$ and (3) $W_{j,j}^{(t)} \leq O(\delta)$;*

2. *(classes already learned) for any class $j \leq i - 1$, we have (1) $b_j^{(t)} \leq \max_{i' \in [k]} b_{i'}^{(t)} - \Omega(1)$, (2) $f_j^{(t)}(x) \geq f_{i'}^{(t)}(x) + \Omega(1)$ for $i' \neq j, x \in \mathcal{S}_j$ and (3) $W_{j,j}^{(t)} \geq \Omega(1)$;*

3. *(parameters movement) (1) for any $j \in [k], \Theta(\delta) = W_{j,j}^{(0)} < W_{j,j}^{(t)}$, (2) for any distinct $j, j' \in [k], 0 < W_{j,j'}^{(t)} \leq O(\delta)$ and (3) for any $j, j' \in [k]$ and any $x \in \mathcal{S}_{j'}, \left| \left\langle W_{j,:}^{(t)}, \xi_x \right\rangle \right| \leq \min\left( O(\delta), W_{j,j'}^{(t)} \right)$.*

This proposition shows that gradient flow learns $k$ classes one by one, from class 1 to class $k$. More precisely, each class $i$ is learned during time $[s_i, s_{i+1}]$. All the not yet learned classes $j \geq i+1$ have close to maximum biases and their weights $W_{j,j}^{(t)}$'s are small. All the already learned classes $j \leq i-1$ have small biases and large weights $W_{j,j}^{(t)}$'s. For the parameters movement, we know that all the diagonal entries $W_{j,j}^{(t)}$'s are larger than the initialization and all the off-diagonal entries $W_{j,j'}^{(t)}$'s are only $O(\delta)$. The correlation between the weights and noise terms also remains small.

When learning class $i$ during time $[s_i, s_{i+1}]$, the weight $W_{i,i}^{(t)}$ slowly grows to a small constant in $[s_i, t_i]$ and then quickly grows large in $[t_i, s_{i+1}.]$ As a result, all $x \in \mathcal{S}_i$ become classified correctly. During the same time, $b_i^{(t)}$ decreases and becomes smaller than the largest bias by at least a constant. At the end time $T = s_k$, although $W_{k,k}^{(T)}$ remains small, all $x \in \mathcal{S}_k$ are also classified correctly because $b_k^{(T)}$ is the largest bias. See an illustration of this learning process in Figure 3.

Although we consider a simple neural network and data distribution, the analysis for the training dynamics is still non-trivial. There are three major challenges in our proof: (1) How to ensure that class $i + 1$ is learned much later than class $i$? (2) For any class $j$ that has not been learned, how to maintain that its bias is close to the maximum? (3) For any learned class $j$, how to maintain the large bias gap from the top bias? Next, we give the proof ideas for these questions. Since all the off-diagonal entries and correlations with noise terms in $W^{(t)}$ are negligible, in our proof we can essentially focus on the movement of $W_{i,i}^{(t)}$'s and $b_i^{(t)}$'s.

**Lower bounding $s_{i+1} - s_i$.** During time $[s_i, t_i]$, the dynamics of $W_{i,i}^{(t)}$ is similar as in the tensor power method (Allen-Zhu & Li, 2020; Ge et al., 2021). The initial gap between $W_{i,i}^{(0)}$ and $W_{j,j}^{(0)}$

ensures that when $W_{i,i}^{(t_i)}$ rises to a small constant, $W_{j,j}^{(t_i)}$ is still $O(\delta)$ for all $j \geq i+1$. Then after constant time $s_{i+1} - t_i$, $W_{j,j}^{s_{i+1}}$ is still $O(\delta)$ since the increasing rate of $W_{j,j}^{(t)}$ is merely $O(\delta^{r-1})$.

**Bias for classes that are not learned.** For $j \geq i+1$, we maintain that $b_j^{(t)} \geq \max_{i' \in [k]} b_{i'}^{(t)} - O(\delta^r)$. First, we use the below lemma to show biases for any two classes $j, j' \geq i+1$ remain close.

**Lemma 1** (Coupling Biases). *Assuming $W_{j',j'}, W_{j,j} \leq O(\delta)$ and $b_{j'}, b_j \geq \max_{i' \in [k]} b_{i'} - O(\delta^r)$, we have $\dot{b}_{j'} - \dot{b}_j > 0$ if $b_{j'} - b_j \leq -\mu\delta^r$, and $\dot{b}_{j'} - \dot{b}_j < 0$ if $b_{j'} - b_j \geq +\mu\delta^r$ for some positive constant $\mu$.*

Second we show that any already learned or being learned class $j' \leq i$ cannot have bias much larger than any class $j \geq i+1$ not yet learned.

**Lemma 2** (Bias Gap Control I). *For any different $j', j \in [k]$, if $W_{j',j'} \geq W_{j,j}, W_{j,j} \leq O(\delta)$ and $b_{j'} - b_j \geq O(\delta^r), b_j \geq \max_{i' \in [k]} b_{i'} - O(\delta^r)$, we have $\dot{b}_{j'} - \dot{b}_j < 0$.*

**Bias for learned classes.** At time $s_{j+1}$, we can prove that $1 - u_j^{(s_{j+1})}(x) \leq C_1$ for all $x \in \mathcal{S}_j$ and $b_j^{(s_{j+1})} - b_k^{(s_{j+1})} \leq -C_2$. According to the below lemma, we can ensure that $b_j^{(t)} - b_k^{(t)} \leq -C_2$ for any $t \geq s_{j+1}$.

**Lemma 3** (Bias Gap Control II). *There exist small positive constants $C_1, C_2$ such that for any $j \in [k-1]$ and any $x \in \mathcal{S}_j$, if $1 - u_j(x) \leq C_1, W_{k,k} \leq O(\delta)$ and $b_j - b_k \geq -C_2$, we have $\dot{b}_j - \dot{b}_k < -\Omega(1)$.*

### 4.1 PLATEAU AND MONOTONICITY FOR $r$-HOMOGENEOUS-WEIGHT NETWORK

Now assuming the network at initialization and after training satisfies the properties described in Theorem 2 and Proposition 1, we can prove a tighter bound on the plateau region and also show the monotonicity in error and loss curve. See the complete proofs in Appendix B.1 and Appendix B.2.

Same as in Assumption 1, we use $\Delta_i$ to denote the bias gap $b_k^{(T)} - b_i^{(T)}$ for $i \in [k-1]$ and denote $\Delta_{\min} := \min_{i \in [k-1]} \Delta_i$ and $\Delta_{\max} = \max_{i \in [k-1]} \Delta_i$. For the weights, we denote $W_{\min} = \min_{i \in [k-1]} W_{i,i}^{(T)}$ and $W_{\max} = \max_{i \in [k]} W_{i,i}^{(T)}$. We denote $R_{\min} = \min_{i \in [k-1]} \Delta_i / [W_{i,i}^{(T)}]^r$, $R_{\max} = \max_{i \in [k-1]} \Delta_i / [W_{i,i}^{(T)}]^r$. Below, we show the plateau and monotonicity of loss and error interpolations in Theorem 3.

**Theorem 3.** *Suppose the neural network, dataset and optimization procedure are as defined in Section 4. Suppose the network at initialization and after training satisfies the properties described in Theorem 2 and Proposition 1. For any $\epsilon \in (0,1)$, suppose $\delta \leq \min(\Theta(\epsilon^{1/r}), \Theta(R_{\min}^{\frac{1}{r-1}} \Delta_{\min}^{1/r}), \Theta((\frac{W_{\min}}{W_{\max}})^{\frac{2r}{r-2}}))$. There exist $\alpha_1 = \frac{\delta}{\Delta_{\min}}, \alpha_2 = (\frac{1}{1+O(\sqrt{\delta})})^{\frac{r}{r-1}} R_{\min}^{\frac{1}{r-1}}, \alpha_3 = \frac{\epsilon^{1/r}}{W_{\max}}$ and $\alpha_4 = (1 + O(\delta))^{\frac{1}{r-1}} \left(\frac{R_{\max}}{r}\right)^{\frac{1}{r-1}}$ such that*

1. *for all $\alpha \in [\alpha_1, \alpha_2]$, the error is $1 - 1/k$; for all $\alpha \in [\alpha_1, 1]$, the error is non-increasing;*

2. *for all $\alpha \in [0, \alpha_3]$, we have $\log k - e\epsilon \leq \frac{1}{N} L(W^{[\alpha]}, b^{[\alpha]}) \leq \log k + \alpha \Delta_{\max} + e\epsilon$; for all $\alpha \in [\alpha_4, 1]$, the loss is strictly monotonically decreasing.*

The analysis for the plateau is very similar as in Theorem 1 since a $r$-homogeneous-weight network is similar to a depth-$r$ fully connected neural network with only last-layer biases in the sense that the weights are $r$-homogeneous while the (last-layer) bias is 1-homogeneous. For the error plateau, we prove a tighter bound on the right boundary $\alpha_2$ than in Theorem 1. We also show the error is non-increasing for $\alpha \in [\delta/\Delta_{\min}, 1]$ by arguing that once a sample is correctly classified at interpolated point $\alpha' \geq \delta/\Delta_{\min}$, it will remain so for any $\alpha \geq \alpha'$. Similar as in Theorem 1, we can show that the loss is no smaller than $\log k - e\epsilon$ when $\alpha \leq \frac{\epsilon^{1/r}}{W_{\max}}$. To show the monotonicity of loss after $\alpha_4$, we show that $f_i^{[\alpha]}(x) - f_j^{[\alpha]}(x)$ is increasing in $\alpha$ for $i \neq j$ and $x \in \mathcal{S}_i$.

In summary, for $\alpha \in [\alpha_1, \alpha_2]$, the signal is smaller than the bias gap and the error remains at $1 - 1/k$. Before $\alpha_3$, the signal is very small and the loss remains large; after $\alpha_4$, the signal starts to overcome the bias gap and the loss is decreasing. See an illustration in Figure 2 (right).

# 5 EXPERIMENTS

In this section we empirically show that intuitions from our simple theoretical model can also be applied to more realistic datasets and architectures. First, we show that bias plays an important role in creating the plateau in the error interpolation, as predicted by Theorem 1. We then demonstrate the influence of initialization size and network depth (also see Theorem 1). Finally, we show that similar to Proposition 1 the class that is learned last often has larger bias. Due to space constraint, we only show the results on MNIST and CIFAR-100 in this section, while similar results also hold on Fashion-MNIST and CIFAR-10 (see Appendix D).

Unless specified otherwise, we use a depth-10 and width-1024 fully-connected ReLU neural network (FCN10) for MNIST and use VGG-16 (without batch normalization) for CIFAR-100. We use Kaiming initialization (He et al., 2015) for the weights and set all bias terms as zeros. For FCN10 on MNIST, we use a small initialization by scaling the weights of each layer by $(0.001)^{1/10}$ so the output is scaled by $0.001$. We train each network using SGD for 100 epochs. See more experiment settings in Appendix D.

We linearly interpolate using 50 evenly spaced points between the network at initialization and the network at the end of training. We evaluate error and loss on the train set. For each setting, we repeat the experiments three times from different random seeds and plot the mean and deviation.

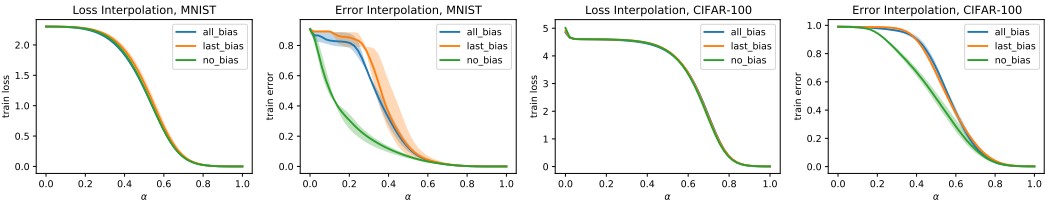

Figure 4: Loss and error curves across networks with all bias, last bias and no bias.

**Role of bias in creating plateau.** We demonstrate the importance of bias using two experiments. In the first experiment, we compare the loss/error interpolation curves between networks equipped with bias for all the layers (*all bias*), with bias only for the output layer (*last bias*), and with no bias at all (*no bias*). Figure 4 shows that networks with all bias and last bias have a much longer error plateau than networks without bias. Three bias settings have similar loss interpolation curves.

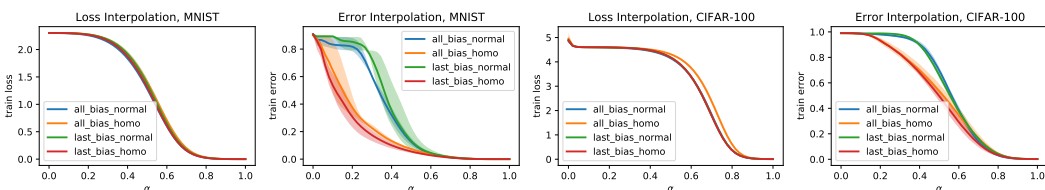

Figure 5: Loss and error curves across networks with normal and homogeneous interpolation on bias.

By our theory, the bias dominates the signal at the beginning of the interpolation because the bias term scales as $\alpha$ while the signal scales as $\alpha^r$. In the second experiment, to correct this discrepancy, we interpolate the bias at the $h$-th layer (input is at the 0-th layer) as $b_h^{[\alpha]} = (1 - \alpha)^h b_h^{(0)} + \alpha^h b_h^{(T)} = \alpha^h b_h^{(T)}$. We call this the *homogeneous interpolation* as now terms involving bias and weights all have $\alpha^r$ coefficients. We compare this with the *normal interpolation* that linearly interpolates the bias terms. Figure 5 shows that for networks with all bias or last bias, using homogeneous interpolation can significantly reduce the plateau in the error interpolation, but does not affect the loss interpolation.

**Role of initialization scale and network depth.** Our theory suggests that with a smaller initialization, the signal magnitude at the initial interpolation is smaller, which can create longer plateau in both loss interpolation and error interpolation. We compare networks under initialization scales $1, 0.1, 0.01$ and $0.001$, where scale 1 corresponds to the standard Kaiming initialization. For other

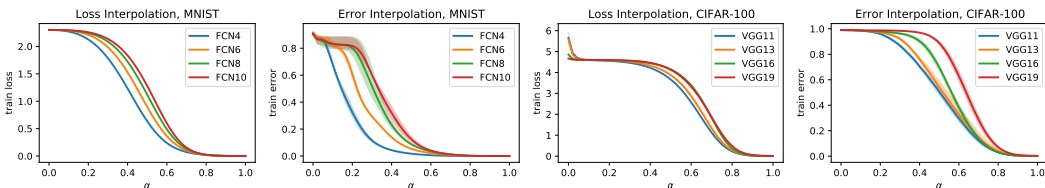

Figure 6: Loss and error curves across networks with different initialization scales.

initialization $\beta$, we rescale each layer by the same factor so the output is rescaled by $\beta$. According to Figure 6, smaller initialization does create longer plateau in loss and error interpolation.

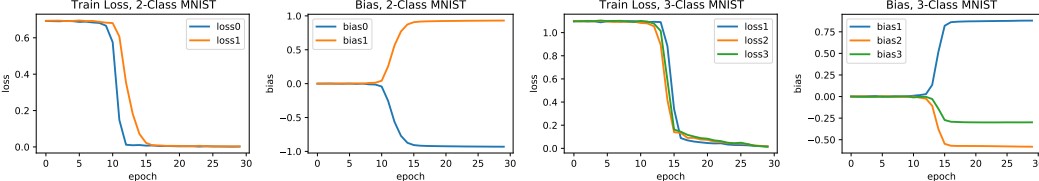

Figure 7: Loss and error curves across networks with different depths.

With a deeper network, the signal grows slower at the initial interpolation phase, which can potentially create a longer plateau in both loss interpolation and error interpolation. We compare FCN4, FCN6, FCN8, FCN10 on MNIST and compare VGG11, VGG13, VGG16, VGG19 on CIFAR-100. According to Figure 7, deeper networks do have longer plateau in loss and error interpolation.

Figure 8: Train loss for each class and bias term dynamics on 2-class MNIST and 3-class MNIST.

**Bias learning dynamics.** Our dynamics analysis in Section 4 shows that gradient descent can learn diverse biases on a balanced dataset by learning different classes at different time points. In particular, the last learned class should have the highest bias term. We verify this theory by studying FCN10 with only output bias on balanced 2-class or 3-class MNIST. To separate the learning of different classes, we compute the per-class loss by only considering the examples in that particular class. According to Figure 8, in the 2-class MNIST, number 1 is learned last and its bias is larger, which fits our theory. Also in the 3-class MNIST, class 2 is learned first, class 3 the second and class 1 the last; for the learned bias, class 2 bias is smallest, class 3 bias in the middle and class 1 bias the highest.

## 6  CONCLUSION

Our theory suggests that the plateau in loss/error interpolation curves may be attributed to simple reasons, and it's unclear if these reasons are related to the difficulty/easiness of optimization. In our experiments although the training succeeds in all the settings, the loss and error interpolation curves can be easily manipulated by changing the initialization size, network depth and bias terms. Therefore, we believe one needs to look at structures more complicated than linear interpolation to understand why optimization succeeds for deep neural networks.

Though our theory requires a small initialization, we also observe plateau in CIFAR-100 with standard initialization, which suggests that the useful signal is still a high order term in $\alpha$. We also observe that sometimes the ordering of the biases does not exactly follow the ordering of the learning. We believe this is partially due to the correlation between different-class features and offer a preliminary explanation in Appendix D.5. We leave the thorough study of these problems in the future work.

## ACKNOWLEDGEMENT

This work is supported by NSF Award DMS-2031849, CCF-1845171 (CAREER), CCF-1934964 (Tripods) and a Sloan Research Fellowship.

## REPRODUCIBILITY STATEMENT

For our theoretical results, we listed all the assumptions and stated the theorems in the main text and we left the complete proof for all the claims in the Appendix. For our experimental results, we defined the detailed experiment settings in the Appendix and also uploaded the source code as supplementary material.

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

# A    EXAMPLES FOR THE DISCONNECTION BETWEEN LINEAR INTERPOLATION SHAPE AND OPTIMIZATION DIFFICULTY

We give two examples that illustrate the disconnection between the linear interpolation shape and the optimization difficulty. In Section A.1, we show a function that is NP-hard to optimize, but has a convex and monotonically decreasing loss interpolation. Then in Section A.2, we give a function that is easy to optimize, but has a non-monotonic loss interpolation.

## A.1    HARD FUNCTION WITH CONVEX LOSS INTERPOLATION

For any symmetric third-order tensor $T \in \mathbb{R}^{d \times d \times d}$, our goal is to minimize

$$f(x, z) = -T(x, x, x) + \|x\|^4 + z^4 \tag{2}$$

where $x \in \mathbb{R}^d$ and $z \in \mathbb{R}$.

It's known that finding the spectral norm of a symmetric third-order tensor (that is, $\max_{v \in \mathbb{R}^d, \|v\|=1} T(v, v, v)$) is NP-hard (Hillar & Lim, 2013). We prove that minimizing $f(x, z)$ is also NP-hard by reducing the tensor spectral norm problem to it.

**Proposition 2.** *Minimizing $f(x, z)$ as defined in Eqn. 2 is NP-hard.*

*Proof.* For any non-zero tensor $T$, let $(x^*, z^*)$ be one minimizer of $f(x, z)$, it's easy to verify that $T(x^*, x^*, x^*) > 0$. We show that $\bar{x}^* := x^* / \|x^*\|$ must be a solution to $\max_{v \in \mathbb{R}^d, \|v\|=1} T(v, v, v)$.

For the sake of contradiction, assume there exists $v^*$ with unit norm such that $T(v^*, v^*, v^*) > T(\bar{x}^*, \bar{x}^*, \bar{x}^*)$. It's easy to verify that $f(\|x^*\| v^*, z^*) < f(x^*, z^*)$, which however contradicts the optimality of $(x^*, z^*)$. $\qquad\square$

Next, we prove that start from certain initialization, the loss along the linear interpolation path is convex and monotonically decreasing. Note that assuming the unit Frobenius norm of $T$ does not hurt the NP-hardness of the problem. And our initialization is oblivious of the tensor $T$.

**Proposition 3.** *Assume $\|T\|_F = 1$. Suppose we start from initialization $(x_0, z_0)$ with $x_0 = 0$ and $|z_0| > \frac{3\sqrt{2}}{4}$. Let $(x^*, z^*)$ be a minimizer of $f(x, z)$ as defined in Eqn. 2. We know the loss interpolation curve $\gamma(\alpha) := f\left((1 - \alpha)x_0 + \alpha x^*, (1 - \alpha)z_0 + \alpha z^*\right)$ is convex and monotonically decreasing for $\alpha \in [0, 1]$.*

*Proof.* We first prove that at any minimizer $(x^*, z^*)$, we must have $z^* = 0$. Otherwise, we can set $z$ as zero to further decrease the loss. Starting from an initialization $(x^{(0)}, z^{(0)})$ with $x^{(0)} = 0$, we know at each interpolation point $x^{[\alpha]} = \alpha x^*, z^{[\alpha]} = (1 - \alpha)z^{(0)}$. Therefore, we have

$$\gamma(\alpha) = f(x^{[\alpha]}, z^{[\alpha]}) = -T(x^{[\alpha]}, x^{[\alpha]}, x^{[\alpha]}) + \left\|x^{[\alpha]}\right\|^4 + \left[z^{[\alpha]}\right]^4$$

$$= -T(\alpha x^*, \alpha x^*, \alpha x^*) + \|\alpha x^*\|^4 + \left[(1 - \alpha)z^{(0)}\right]^4$$

$$= -\alpha^3 T(x^*, x^*, x^*) + \alpha^4 \|x^*\|^4 + (1 - \alpha)^4 \left[z^{(0)}\right]^4.$$

To prove the convexity of $\gamma(\alpha)$ for $\alpha \in [0, 1]$, we only need to prove $\gamma''(\alpha) > 0$ for $\alpha \in [0, 1]$. We have

$$\gamma''(\alpha) = -6\alpha T(x^*, x^*, x^*) + 12\alpha^2 \|x^*\|^4 + 12(1 - \alpha)^2 \left[z^{(0)}\right]^4$$

$$= -6\alpha \|x^*\|^3 T(\bar{x}^*, \bar{x}^*, \bar{x}^*) + 12\alpha^2 \|x^*\|^4 + 12(1 - \alpha)^2 \left[z^{(0)}\right]^4.$$

Since the formula for $\gamma''(\alpha)$ involves both $T(\bar{x}^*, \bar{x}^*, \bar{x}^*)$ and $\|x^*\|$, we first figure out the relation between these two quantities. Suppose $T(\bar{x}^*, \bar{x}^*, \bar{x}^*) = p > 0$, it's not hard to find $\|x^*\|$ must be equal to $\frac{3p}{4}$. This is because $-\|x^*\|^3 p + \|x^*\|^4$ is minimized when $\|x^*\| = \frac{3p}{4}$. Next, we prove $\gamma''(\alpha) > 0$ for $\alpha \in (2/3, 1]$ and $\alpha \in [0, 2/3]$ seperately.

When $\alpha \in (2/3, 1]$, we have

$$12\alpha^2 \left\| x^* \right\|^4 > 6p\alpha \left\| x^* \right\|^3 = 6\alpha \left\| x^* \right\|^3 T(\bar{x}^*, \bar{x}^*, \bar{x}^*).$$

Therefore, we know $\gamma''(\alpha) > 0$.

When $\alpha \in [0, 2/3]$, we know

$$12(1-\alpha)^2 \left[ z^{(0)} \right]^4 \geq \frac{4}{3} \left[ z^{(0)} \right]^4.$$

Since $\|T\|_F = 1$, we know $T(\bar{x}^*, \bar{x}^*, \bar{x}^*) \leq 1$ and $\|x^*\| \leq 3/4$. Therefore, we have

$$6\alpha \left\| x^* \right\|^3 T(\bar{x}^*, \bar{x}^*, \bar{x}^*) \leq 6 \cdot \frac{2}{3} \cdot \left( \frac{3}{4} \right)^3 \cdot 1 = \frac{27}{16}.$$

Then, we know that if $\left| z^{(0)} \right| > \frac{3\sqrt{2}}{4}$, we have $\gamma''(\alpha) > 0$. $\qquad\square$

## A.2 Easy function with non-monotonic loss interpolation

In this section, we give an easy-to-optimize function that however has a non-monotonic loss interpolation curve. We consider the following loss function

$$f(x, y) = \begin{cases} 0 & \text{if } x = y = 0 \\ \left( 1 - \frac{y}{3\sqrt{x^2+y^2}} \right) \left( \left( x^2 + y^2 \right)^2 - 2 \left( x^2 + y^2 \right) \right) & \text{otherwise}, \end{cases} \tag{3}$$

where $x, y \in \mathbb{R}$. We can also re-parameterize $f(x, y)$ using angle $\theta \in [0, 2\pi)$ and length $r \in [0, \infty)$ as $h(\theta, r) = \left( 1 - \frac{\sin(\theta)}{3} \right) \left( r^4 - 2r^2 \right)$.

Next, we prove that starting from any non-zero point, gradient flow converges to the global minimizer.

**Proposition 4.** *Starting from any non-zero initialization, gradient flow on $f(x, y)$ as defined in Eqn. 3 converges to the global minimizer $(0, -1)$.*

*Proof.* We know the unique minimizer of $f(x, y)$ is $(0, -1)$ by considering its equivalent form $h(\theta, r) = \left( 1 - \frac{\sin(\theta)}{3} \right) \left( r^4 - 2r^2 \right)$. For $h(\theta, r) = \left( 1 - \frac{\sin(\theta)}{3} \right) \left( r^4 - 2r^2 \right)$, we know $\left( r^4 - 2r^2 \right)$ is minimized at $r = 1$ and $\left( 1 - \frac{\sin(\theta)}{3} \right)$ is maximized at $\theta = \frac{3\pi}{2}$.

Besides the minimizer $(0, -1)$, the other stationary point is at $(0, 0)$. For any point $(x, y)$ different from $(0, -1)$ and $(0, 0)$, if $x^2 + y^2 \neq 1$, the gradient along the radial direction is non-zero; if $\frac{y}{\sqrt{x^2+y^2}} \neq -1$, the gradient along the tangent direction is non-zero. It's also easy to verify that starting from a non-zero point, gradient flow does not converge to $(0, 0)$, so it must converge to $(0, -1)$ $\qquad\square$

It's also very easy to prove that gradient descent with appropriate step size converges to an $\epsilon$-neighborhood of the global minimizer within $\text{poly}(1/\epsilon)$ number of iterations. This is because the gradient is at least $\text{poly}(\epsilon)$ for any non-zero point outside of the $\epsilon$-neighborhood of the global minimizer. Starting from an initialization $(x, y)$ with $x^2 + y^2 = \Theta(1)$, the smoothness along the training is also bounded by a constant.

Next, we prove that starting from certain initialization [3], the loss interpolation between the initialization and the global minimizer is non-monotonic. We prove this by identifying two points along the interpolation path such that the point closer to minimizer has a higher loss compared with the point further to the minimizer.

---

[3]Note the initialization condition in Prop. 5 is satisfied with constant probability for a reasonable initialization scheme. For example, if we uniformly sample $(x, y)$ from the set $S = \{(x, y) \in \mathbb{R}^2 | x^2 + y^2 \leq R\}$ with $R \geq 2$, the condition is satisfied with constant probability.

**Proposition 5.** *Suppose we start from an initialization* $(x_0, y_0) = (r\sin(\beta), r\cos(\beta))$ *with* $r \geq 1$ *and* $\beta \in [-\pi/3, \pi/3]$. *Consider the loss interpolation curve* $\gamma(\alpha) = f((1-\alpha)x_0 + \alpha x^*, (1-\alpha)y_0 + \alpha y^*)$ *with* $(x^*, y^*) = (0, -1)$ *and* $f(\cdot, \cdot)$ *defined in Eqn. 3. We know there exist* $0 \leq \alpha_1 < \alpha_2 \leq 1$ *such that*

$$\gamma(\alpha_2) - \gamma(\alpha_1) \geq \frac{5}{32}.$$

*Proof.* We prove for any $\beta \in [-\pi/3, \pi/3]$ and any $r \geq 1$, the loss interpolation between $(r\sin(\beta), r\cos(\beta))$ to $(0, -1)$ is non-monotonic. In particular, we show there are two points along the linear interpolation satisfying

$$f\left(\sin\left(\beta/2\right)\cos\left(\beta/2\right), -\sin\left(\beta/2\right)\sin\left(\beta/2\right)\right) - f\left(\sin(\beta), \cos(\beta)\right) \geq 1/12,$$

where $(\sin\left(\beta/2\right)\cos\left(\beta/2\right), -\sin\left(\beta/2\right)\sin\left(\beta/2\right))$ is the middle point between $(\sin(\beta), \cos(\beta))$ and $(0, -1)$.

Next, we separately upper bound $f\left(\sin(\beta), \cos(\beta)\right)$ and lower bound $f\left(\sin\left(\beta/2\right)\cos\left(\beta/2\right), -\sin\left(\beta/2\right)\sin\left(\beta/2\right)\right)$. We have

$$\max_{\beta \in [-\pi/3, \pi/3]} f\left(\sin(\beta), \cos(\beta)\right) \leq f(0, 1) = -\frac{2}{3}$$

and

$$\min_{\beta \in [-\pi/3, \pi/3]} f\left(\sin\left(\beta/2\right)\cos\left(\beta/2\right), -\sin\left(\beta/2\right)\sin\left(\beta/2\right)\right)$$
$$\geq f\left(\sin\left(\pi/6\right)\cos\left(\pi/6\right), -\sin\left(\pi/6\right)\sin\left(\pi/6\right)\right)$$
$$= \left(1 + \frac{1}{2} \cdot \frac{1}{3}\right)\left(\left(\frac{1}{2}\right)^4 - 2\left(\frac{1}{2}\right)^2\right)$$
$$= -\frac{49}{96}.$$

Therefore, we have $f\left(\sin\left(\beta/2\right)\cos\left(\beta/2\right), -\sin\left(\beta/2\right)\sin\left(\beta/2\right)\right) - f\left(\sin(\beta), \cos(\beta)\right) \geq \frac{5}{32}$. $\square$

# B    PROOF FOR PLATEAU AND MONOTONICITY

We first consider the $r$-homogeneous-weight model. We prove the plateau and monotonicity properties for the error interpolation (Theorem 4) in Section B.1. We then prove the plateau and monotonicity properties for the loss interpolation (Theorem 5) in Section B.2. Theorem 3 is a simple combination of Theorem 4 and Theorem 5. Finally, we give the plateau analysis for the fully-connected neural networks (Theorem 1) in Section B.3.

## B.1    ERROR INTERPOLATION FOR $r$-HOMOGENEOUS-WEIGHT MODEL

**Theorem 4** (Error Interpolation). *Suppose the network at initialization and after training satisfy the properties described in Theorem 2 and Induction Hypothesis 1. Suppose* $\delta \leq \min(O(1), O(R_{\min}^{\frac{1}{r-1}}\Delta_{\min}^{1/r}), O((\frac{W_{\min}}{W_{\max}})^{\frac{2r}{r-2}}))$. *There exist* $\alpha_1 = \frac{\delta}{\Delta_{\min}}$ *and* $\alpha_2 = \left(\frac{1}{1+O(\sqrt{\delta})}\right)^{\frac{r}{r-1}}R_{\min}^{\frac{1}{r-1}}$, *such that*

1. *for all* $\alpha \in [\alpha_1, \alpha_2]$, *the error is* $1 - 1/k$;

2. *for all* $\alpha \in [\alpha_1, 1]$, *the error is non-increasing.*

**Proof of Theorem 4.** This theorem directly follows from Lemma 4 and Lemma 5. $\square$

Next, we separately prove the initial plateau in Lemma 4 and the monotonicity in Lemma 5.

**Lemma 4** (Error Plateau). *In the same setting as in Theorem 4, there exists* $\alpha_1 = \frac{\delta}{\Delta_{\min}}$ *and* $\alpha_2 = \left(\frac{1}{1+O(\sqrt{\delta})}\right)^{\frac{r}{r-1}}R_{\min}^{\frac{1}{r-1}}$, *such that for any interpolation point with* $\alpha \in [\alpha_1, \alpha_2]$, *the error is* $1 - 1/k$. *Moreover, we have* $f_i^{[\alpha]}(e_j) < f_k^{[\alpha]}(e_j)$ *for all* $j \in [k]$ *and all* $i \neq k$.

In the proof of Lemma 4, we show that for interpolation point $\alpha \in [\alpha_1, \alpha_2]$, the bias term dominates and all samples are classified as class $k$ that has the largest bias.

**Proof of Lemma 4.** We only need to show that for all $\alpha \in [\alpha_1, \alpha_2]$, we have

$$f_i^{[\alpha]}(x) < f_k^{[\alpha]}(x)$$

for all $x \in \mathcal{S}$ and all $i \neq k$, which immediately implies the error is $1 - 1/k$. Without loss of generality, assume $x \in \mathcal{S}_j$ where $j$ may equal $i$ or $k$.

**For $\alpha \in \left[\alpha_1, \frac{\sqrt{\delta}}{W_{\min}}\right)$.** If $\alpha_1 = \frac{\delta}{\Delta_{\min}} \geq \frac{\sqrt{\delta}}{W_{\min}}$, we only need to consider the case when $\alpha \in \left[\frac{\sqrt{\delta}}{W_{\min}}, \alpha_2\right]$. So here we assume $\frac{\delta}{\Delta_{\min}} < \frac{\sqrt{\delta}}{W_{\min}}$. We can lower bound $f_k^{[\alpha]}(x) - f_i^{[\alpha]}(x)$ as

$$
\begin{aligned}
&f_k^{[\alpha]}(x) - f_i^{[\alpha]}(x) \\
&= \left[\left\langle W_{k,:}^{[\alpha]}, x\right\rangle\right]^r + b_k^{[\alpha]} - \left[\left\langle W_{i,:}^{[\alpha]}, x\right\rangle\right]^r - b_i^{[\alpha]} \\
&= \left[\left\langle W_{k,:}^{[\alpha]}, x\right\rangle\right]^r + b_k^{[\alpha]} - \left[W_{i,j}^{[\alpha]} \pm O(\delta)\right]^r - b_i^{[\alpha]} \\
&\geq \alpha \Delta_i - \left[W_{i,j}^{(0)} + \alpha W_{i,j}^{(T)} + O(\delta)\right]^r,
\end{aligned}
$$

where the second equality uses $\left|\left\langle W_{i,:}^{[\alpha]}, \xi_x\right\rangle\right| \leq O(\delta)$ and the inequality uses $\left\langle W_{k,:}^{[\alpha]}, x\right\rangle \geq 0$.

To prove $f_k^{[\alpha]}(x) - f_i^{[\alpha]}(x) > 0$ for $\alpha \in \left[\frac{\delta}{\Delta_{\min}}, \frac{\sqrt{\delta}}{W_{\min}}\right]$, we only need to prove $\frac{\delta}{\Delta_{\min}} \Delta_i - \left[W_{i,j}^{(0)} + \frac{\sqrt{\delta}}{W_{\min}} W_{i,j}^{(T)} + O(\delta)\right]^r > 0$. Since $\Delta_i \geq \Delta_{\min}$, we know $\frac{\delta}{\Delta_{\min}} \Delta_i \geq \delta$. Due to full accuracy, we know $\left\langle W_{i,:}^{(T)}, x\right\rangle \geq \Delta_i^{1/r}$ for $x \in \mathcal{S}_i$, which then implies $W_{i,i}^{(T)} \geq \Omega\left(\Delta_i^{1/r}\right)$ because $\Delta_i \geq \Omega(1)$ and $\left\langle W_{i,:}^{(T)}, \xi_x\right\rangle \leq O(\delta) \leq O(1)$. Since $W_{i,i}^{(T)} \geq \Omega\left(\Delta_i^{1/r}\right)$ and $W_{i,j}^{(T)} \leq O(\delta)$ for $i \neq j$, so we have $W_{i,j}^{(T)} \leq W_{i,i}^{(T)} \leq W_{\max}$ as long as $\delta \leq O(\Delta_{\min}^{1/r})$. So we can upper bound $\left[W_{i,j}^{(0)} + \frac{\sqrt{\delta}}{W_{\min}} W_{i,j}^{(T)} + O(\delta)\right]^r$ as follows,

$$
\begin{aligned}
\left[W_{i,j}^{(0)} + \frac{\sqrt{\delta}}{W_{\min}} W_{i,j}^{(T)} + O(\delta)\right]^r &\leq \left[O(\delta) + \frac{\sqrt{\delta} W_{\max}}{W_{\min}}\right]^r \\
&\leq \left[\frac{\sqrt{\delta} W_{\max}}{r W_{\min}} + \frac{\sqrt{\delta} W_{\max}}{W_{\min}}\right]^r \\
&\leq e \left(\frac{\sqrt{\delta} W_{\max}}{W_{\min}}\right)^r,
\end{aligned}
$$

where the second inequality assumes $\delta \leq O\left(\frac{W_{\max}^2}{W_{\min}^2}\right)$. Therefore, to prove $\frac{\delta}{\Delta_{\min}} \Delta_i - \left[W_{i,j}^{(0)} + \frac{\sqrt{\delta}}{W_{\min}} W_{i,j}^{(T)} + O(\delta)\right]^r > 0$ we only need

$$\delta - e\left(\frac{\sqrt{\delta} W_{\max}}{W_{\min}}\right)^r > 0,$$

which holds as long as $\delta < \left[\frac{1}{e}\left(\frac{W_{\min}}{W_{\max}}\right)^r\right]^{\frac{2}{r-2}}$.

**For $\alpha \in \left[\frac{\sqrt{\delta}}{W_{\min}}, \alpha_2\right]$.** Similar as above, we only need to show that $\alpha \Delta_i - \left[W_{i,j}^{(0)} + \alpha W_{i,j}^{(T)} + O(\delta)\right]^r > 0$ for $i \neq k$ and $j \in [k]$. Since $W_{i,j}^{(0)} \leq O(\delta)$ and $\alpha \geq \sqrt{\delta}/W_{\min}$, we

have $W_{i,j}^{(0)} \leq O(\sqrt{\delta}\alpha W_{\min})$. Therefore, we have $W_{i,j}^{(0)} + \alpha W_{i,j}^{(T)} + O(\delta) \leq \left(1 + O(\sqrt{\delta})\right)\alpha W_{i,i}^{(T)}$. Therefore, we have

$$\alpha \Delta_i - \left[W_{i,j}^{(0)} + \alpha W_{i,j}^{(T)} + O(\delta)\right]^r \geq \alpha \Delta_i - \left(1 + O(\sqrt{\delta})\right)^r \alpha^r \left[W_{i,i}^{(T)}\right]^r > 0,$$

where the last inequality assumes $\alpha \leq \alpha_2 := \left(\frac{1}{1+O(\sqrt{\delta})}\right)^{\frac{r}{r-1}} R_{\min}^{\frac{1}{r-1}}$ where $R_{\min} = \min_{i \in [k-1]} \Delta_i / [W_{i,i}^{(T)}]^r$. □

Next, we show that the error is non-increasing for $\alpha \in [\alpha_1, 1]$ by proving that once a sample is classified correctly it will remain so.

**Lemma 5** (Error Monotonicity). *In the same setting as in Theorem 4, there exists $\delta_1 = \frac{\delta}{\Delta_{\min}}$ such that the error is non-increasing for $\alpha \in [\alpha_1, 1]$.*

**Proof of Lemma 5.** We first show that sample $e_k$ is correctly classified for the whole range $[\alpha_1, 1]$. Second, we show for any other sample once it become classified right it will remain so. Combining these two cases, we prove the monotonicity of the error rate.

**Class k.** We first show that every $x \in S_k$ is classified correctly for any $\alpha \in [\alpha_1, 1]$. According to Lemma 4, we know that

$$f_k^{[\alpha_1]}(x) > f_i^{[\alpha_1]}(x)$$

for any $i \neq k$. We only need to prove that $f_k^{[\alpha]}(x) - f_i^{[\alpha]}(x)$ is increasing for $\alpha \in [\alpha_1, 1]$. Expanding $f_k^{[\alpha]}(x) - f_i^{[\alpha]}(x)$, we have

$$f_k^{[\alpha]}(x) - f_i^{[\alpha]}(x)$$
$$= \left[(1-\alpha)\left\langle W_{k,:}^{(0)}, x\right\rangle + \alpha\left(\left\langle W_{k,:}^{(T)}, x\right\rangle\right)\right]^r - \left[(1-\alpha)\left\langle W_{i,:}^{(0)}, x\right\rangle + \alpha\left\langle W_{i,:}^{(T)}, x\right\rangle\right]^r + \alpha\left(b_k^{(T)} - b_i^{(T)}\right),$$

which is increasing since $\left|\left\langle W_{k,:}^{(T)}, x\right\rangle\right|, \left|\left\langle W_{k,:}^{(0)}, x\right\rangle\right|, \left|\left\langle W_{i,:}^{(T)}, x\right\rangle\right|, \left|\left\langle W_{i,:}^{(0)}, x\right\rangle\right| \leq O(\delta)$ and $b_k^{(T)} - b_i^{(T)} > \Omega(1)$.

**Other classes.** For any class $i \neq k$, from Lemma 4, we know that it is classified incorrectly for $\alpha \in [\alpha_1, \alpha_2]$. We prove that once it become classified correctly at some $\alpha' \in (\alpha_2, 1]$, it remains so for $\alpha \in [\alpha', 1]$.

We show that at $\alpha$, for any $x \in S_i$, if $f_i^{[\alpha]}(x) > f_j^{[\alpha]}(x)$ for all $j \neq i$, we have $\frac{\partial}{\partial \alpha}\left(f_i^{[\alpha]}(x) - f_j^{[\alpha]}(x)\right) > 0$. Expanding $f_i^{[\alpha]}(x) - f_j^{[\alpha]}(x)$, we have

$$f_i^{[\alpha]}(x) - f_j^{[\alpha]}(x)$$
$$= \left[\left\langle W_{i,:}^{[\alpha]}, x\right\rangle\right]^r + b_i^{[\alpha]} - \left[\left\langle W_{j,:}^{[\alpha]}, x\right\rangle\right]^r - b_j^{[\alpha]}$$
$$= \left[\left\langle W_{i,:}^{[\alpha]}, x\right\rangle\right]^r - \left[\left\langle W_{j,:}^{[\alpha]}, x\right\rangle\right]^r - \alpha\left(b_j^{(T)} - b_i^{(T)}\right).$$

Since $f_i^{[\alpha]}(x) - f_j^{[\alpha]}(x) > 0$, we have

$$\left[\left\langle W_{i,:}^{[\alpha]}, x\right\rangle\right]^r > \alpha\left(b_j^{(T)} - b_i^{(T)}\right),$$

where we use $\left\langle W_{j,:}^{[\alpha]}, x\right\rangle \geq 0$. Computing $\frac{\partial}{\partial \alpha}\left(f_i^{[\alpha]}(x) - f_j^{[\alpha]}(x)\right)$, we have

$$\frac{\partial}{\partial \alpha}\left(f_i^{[\alpha]}(x) - f_j^{[\alpha]}(x)\right)$$
$$= \frac{\partial}{\partial \alpha}\left(\left[(1-\alpha)\left\langle W_{i,:}^{(0)}, x\right\rangle + \alpha\left\langle W_{i,:}^{(T)}, x\right\rangle\right]^r - \left[(1-\alpha)\left\langle W_{j,:}^{(0)}, x\right\rangle + \alpha\left\langle W_{j,:}^{(T)}, x\right\rangle\right]^r + \alpha\left(b_i^{(T)} - b_j^{(T)}\right)\right)$$
$$\geq r\left[\left\langle W_{i,:}^{(0)}, x\right\rangle + \alpha\left(\left\langle W_{i,:}^{(T)}, x\right\rangle - \left\langle W_{i,:}^{(0)}, x\right\rangle\right)\right]^{r-1}\left(\left\langle W_{i,:}^{(T)}, x\right\rangle - \left\langle W_{i,:}^{(0)}, x\right\rangle\right) - \left(b_j^{(T)} - b_i^{(T)}\right) - O(\delta^r),$$

where the inequality uses $\left|\left\langle W_{j,:}^{(0)}, x\right\rangle\right|, \left|\left\langle W_{j,:}^{(T)}, x\right\rangle\right| \le O(\delta)$.

If $b_j^{(T)} - b_i^{(T)} \le 0$, we only need to prove

$$r\left[\left\langle W_{i,:}^{(0)}, x\right\rangle + \alpha\left(\left\langle W_{i,:}^{(T)}, x\right\rangle - \left\langle W_{i,:}^{(0)}, x\right\rangle\right)\right]^{r-1}\left(\left\langle W_{i,:}^{(T)}, x\right\rangle - \left\langle W_{i,:}^{(0)}, x\right\rangle\right) - O(\delta^r) > 0,$$

which holds since $\left(\left\langle W_{i,:}^{(T)}, x\right\rangle - \left\langle W_{i,:}^{(0)}, x\right\rangle\right), \left\langle W_{i,:}^{[\alpha]}, x\right\rangle \ge \Omega(1)$.

If $b_j^{(T)} - b_i^{(T)} > 0$, we have

$$\frac{\partial}{\partial\alpha}\left(f_i^{[\alpha]}(x) - f_j^{[\alpha]}(x)\right)$$

$$= r\left[\left\langle W_{i,:}^{(0)}, x\right\rangle + \alpha\left(\left\langle W_{i,:}^{(T)}, x\right\rangle - \left\langle W_{i,:}^{(0)}, x\right\rangle\right)\right]^{r-1}\left(\left\langle W_{i,:}^{(T)}, x\right\rangle - \left\langle W_{i,:}^{(0)}, x\right\rangle\right) - \left(b_j^{(T)} - b_i^{(T)}\right) - O(\delta^r)$$

$$> \frac{(1 - O(\delta^r))\, r\left(\left\langle W_{i,:}^{(T)}, x\right\rangle - \left\langle W_{i,:}^{(0)}, x\right\rangle\right)}{\left[\left\langle W_{i,:}^{(0)}, x\right\rangle + \alpha\left(\left\langle W_{i,:}^{(T)}, x\right\rangle - \left\langle W_{i,:}^{(0)}, x\right\rangle\right)\right]} \cdot \alpha\left(b_j^{(T)} - b_i^{(T)}\right) - \left(b_j^{(T)} - b_i^{(T)}\right),$$

where the last inequality uses $\left[\left\langle W_{i,:}^{[\alpha]}, x\right\rangle\right]^r > \alpha\left(b_j^{(T)} - b_i^{(T)}\right)$. Therefore, to prove $\frac{\partial}{\partial\alpha}\left(f_i^{[\alpha]}(e_i) - f_j^{[\alpha]}(e_i)\right) > 0$, we only need to prove $\frac{(1-O(\delta^r))r\left(\left\langle W_{i,:}^{(T)},x\right\rangle-\left\langle W_{i,:}^{(0)},x\right\rangle\right)}{\left[\left\langle W_{i,:}^{(0)},x\right\rangle+\alpha\left(\left\langle W_{i,:}^{(T)},x\right\rangle-\left\langle W_{i,:}^{(0)},x\right\rangle\right)\right]} \ge \frac{1}{\alpha}$.
We have

$$\frac{(1 - O(\delta^r))\, r\left(\left\langle W_{i,:}^{(T)}, x\right\rangle - \left\langle W_{i,:}^{(0)}, x\right\rangle\right)}{\left[\left\langle W_{i,:}^{(0)}, x\right\rangle + \alpha\left(\left\langle W_{i,:}^{(T)}, x\right\rangle - \left\langle W_{i,:}^{(0)}, x\right\rangle\right)\right]} \ge \frac{(1 - O(\delta^r))\, r\left(\left\langle W_{i,:}^{(T)}, x\right\rangle - \left\langle W_{i,:}^{(0)}, x\right\rangle\right)}{2\alpha\left(\left\langle W_{i,:}^{(T)}, x\right\rangle - \left\langle W_{i,:}^{(0)}, x\right\rangle\right)} \ge \frac{1}{\alpha}.$$

The first inequality requires $\left\langle W_{i,:}^{(0)}, x\right\rangle \le \alpha\left(\left\langle W_{i,:}^{(T)}, x\right\rangle - \left\langle W_{i,:}^{(0)}, x\right\rangle\right)$ and the second inequality uses $r \ge 3, (1 - O(\delta^r)) \ge 2/3$. To prove $\left\langle W_{i,:}^{(0)}, x\right\rangle \le \alpha\left(\left\langle W_{i,:}^{(T)}, x\right\rangle - \left\langle W_{i,:}^{(0)}, x\right\rangle\right)$, it's equivalent to show $\left\langle W_{i,:}^{(0)}, x\right\rangle \le \frac{\alpha}{1+\alpha}\left\langle W_{i,:}^{(T)}, x\right\rangle$. Since $\alpha \ge \alpha_2 = \left(\frac{1}{1+O(\sqrt{\delta})}\right)^{\frac{r}{r-1}} R_{\min}^{\frac{1}{r-1}}$, we can lower bound $\frac{\alpha}{1+\alpha}$ as follows,

$$\frac{\alpha}{1 + \alpha} \ge \frac{1}{2}\left(\frac{1}{1 + O(\sqrt{\delta})}\right)^{\frac{r}{r-1}} R_{\min}^{\frac{1}{r-1}}$$

$$\ge \frac{1}{8} R_{\min}^{\frac{1}{r-1}},$$

where the first inequality uses $\alpha \le 1$ and the second inequality uses $1 + O(\sqrt{\delta}) \le 2, r \ge 2$. So we have $\frac{\alpha}{1+\alpha}\left\langle W_{i,:}^{(T)}, x\right\rangle \ge \frac{1}{8} R_{\min}^{\frac{1}{r-1}} \Delta_{\min}^{1/r}$. Therefore, we only need $\delta \le O\left(R_{\min}^{\frac{1}{r-1}} \Delta_{\min}^{1/r}\right)$ to ensure that $\left\langle W_{i,:}^{(0)}, x\right\rangle \le \alpha\left(\left\langle W_{i,:}^{(T)}, x\right\rangle - \left\langle W_{i,:}^{(0)}, x\right\rangle\right)$. $\qquad\square$

## B.2 LOSS INTERPOLATION FOR $r$-HOMOGENEOUS-WEIGHT MODEL

In this section, we give a proof of Theorem 5.

**Theorem 5** (Loss Interpolation). *Suppose the network at initialization and after training satisfy the properties described in Theorem 2 and Induction Hypothesis 1. For any $\epsilon \in (0, 1)$, suppose $\delta \le O(\epsilon^{1/r})$, there exist $\alpha_3 = \frac{\epsilon^{1/r}}{W_{\max}}$ and $\alpha_4 = (1 + O(\delta))^{\frac{1}{r-1}}\left(\frac{R_{\max}}{r}\right)^{\frac{1}{r-1}}$ such that*

*1. for all $\alpha \in [0, \alpha_3]$, we have $\log k - e\epsilon \le \frac{1}{N} L(W^{[\alpha]}, b^{[\alpha]}) \le \log k + \alpha\Delta_{\max} + e\epsilon$;*

*2. for all $\alpha \in [\alpha_4, 1]$, the loss is monotonically decreasing.*

**Proof of Theorem 5.** This theorem directly follows from Lemma 6 and Lemma 7. $\qquad\square$

Next, we prove the initial loss plateau in Lemma 6 and the monotonicity in Lemma 7.

**Lemma 6** (Loss Plateau). *In the same setting as in Theorem 5, for any $\epsilon > 0$, there exists $\alpha_3 = \frac{\epsilon^{1/r}}{W_{\max}}$ such that for all $\alpha \in [0, \alpha_3]$*

$$N\left(\log k - e\epsilon\right) \leq L(W^{[\alpha]}, b^{[\alpha]}) \leq N\left(\log k + \alpha\Delta_{\max} + e\epsilon\right).$$

We show that for $\alpha \in [0, \alpha_3]$, the weights $W^{[\alpha]}$ is negligible and the bias dominates, which then gives a lower bound and an upper bound of the loss.

**Proof of Lemma 6.** Since $\alpha \leq \alpha_3 = \frac{\epsilon^{1/r}}{W_{\max}}$ and $\delta \leq O\left(\epsilon^{1/r}\right)$, we have

$$\left[\left\langle W_{i,:}^{[\alpha]}, x\right\rangle\right]^r = \left[\left\langle W_{i,:}^{(0)}, x\right\rangle + \alpha(\left\langle W_{i,:}^{(T)}, x\right\rangle - \left\langle W_{i,:}^{(0)}, x\right\rangle)\right]^r \leq \left[\left(1 + \frac{1}{r}\right)\epsilon^{1/r}\right]^r \leq e\epsilon,$$

for all $i \in [k], x \in \mathcal{S}$.

We can divide the dataset $\mathcal{S}$ into $N/k$ disjoint subsets $\{P_l\}_{l=1}^{N/k}$ where each $P_l$ contains exactly one sample from each class. Next, we bound the total loss of each subset $P_l$. Without loss of generality, let's consider subset $P_1$ and suppose $x^{(i)}$ is the $i$-th class sample in this subset. For convenience, we denote the total loss of samples in $P_1$ as $L_1(W^{[\alpha]}, b^{[\alpha]})$.

**Lower bounding $L_1(W^{[\alpha]}, b^{[\alpha]})$.** We have

$$
\begin{aligned}
L_1(W^{[\alpha]}, b^{[\alpha]}) &= \sum_{i\in[k]} \log\left(\frac{\sum_{j\in[k]} \exp\left(f_j^{[\alpha]}(x^{(i)})\right)}{\exp\left(f_i^{[\alpha]}(x^{(i)})\right)}\right) \\
&= \log\left(\prod_{i\in[k]} \frac{\sum_{j\in[k]} \exp\left(f_j^{[\alpha]}(x^{(i)})\right)}{\exp\left(f_i^{[\alpha]}(x^{(i)})\right)}\right) \\
&\geq \log\left(\left(\frac{k}{\sum_{i\in[k]} \frac{\exp\left(f_i^{[\alpha]}(x^{(i)})\right)}{\sum_{j\in[k]} \exp\left(f_j^{[\alpha]}(x^{(i)})\right)}}\right)^k\right),
\end{aligned}
$$

where the last inequality uses the HM-GM inequality. We can then upper bound $\sum_{i\in[k]} \frac{\exp\left(f_i^{[\alpha]}(x^{(i)})\right)}{\sum_{j\in[k]} \exp\left(f_j^{[\alpha]}(x^{(i)})\right)}$ as follows,

$$
\begin{aligned}
\sum_{i\in[k]} \frac{\exp\left(f_i^{[\alpha]}(x^{(i)})\right)}{\sum_{j\in[k]} \exp\left(f_j^{[\alpha]}(x^{(i)})\right)} &= \sum_{i\in[k]} \frac{\exp\left(\left[\left\langle W_{i,:}^{[\alpha]}, x^{(i)}\right\rangle\right]^r + \alpha b_i\right)}{\sum_{j\in[k]} \exp\left(\left[\left\langle W_{j,:}^{[\alpha]}, x^{(i)}\right\rangle\right]^r + \alpha b_j\right)} \\
&\leq \sum_{i\in[k]} \frac{\exp\left(\alpha b_i + e\epsilon\right)}{\sum_{j\in[k]} \exp\left(\alpha b_j\right)} \\
&= \frac{\sum_{i\in[k]} \exp\left(\alpha b_i\right)}{\sum_{j\in[k]} \exp\left(\alpha b_j\right)} \exp\left(e\epsilon\right) \\
&= \exp\left(e\epsilon\right).
\end{aligned}
$$

Plugging back to the lower bound of $L_1(W^{[\alpha]}, b^{[\alpha]})$, we have

$$L_1(W^{[\alpha]}, b^{[\alpha]}) \geq k\log\left(\frac{k}{\exp\left(e\epsilon\right)}\right) = k\left(\log k - e\epsilon\right).$$

**Upper bounding $L_1(W^{[\alpha]}, b^{[\alpha]})$.** We have

$$
\begin{aligned}
L_1(W^{[\alpha]}, b^{[\alpha]}) &= \sum_{i \in [k]} \log \left( \frac{\sum_{j \in [k]} \exp \left( f_j^{[\alpha]}(x^{(i)}) \right)}{\exp \left( f_i^{[\alpha]}(x^{(i)}) \right)} \right) \\
&\leq \sum_{i \in [k]} \log \left( \frac{\sum_{j \in [k]} \exp \left( \alpha b_j + e\epsilon \right)}{\exp \left( \alpha b_i \right)} \right) \\
&\leq k \log \left( k \exp \left( \alpha \Delta_{\max} + e\epsilon \right) \right) \\
&\leq k \left( \log k + \alpha \Delta_{\max} + e\epsilon \right)
\end{aligned}
$$

The above analysis applies for every subset $P_l$, so we have

$$
N \left( \log k - e\epsilon \right) \leq L(W^{[\alpha]}, b^{[\alpha]}) \leq N \left( \log k + \alpha \Delta_{\max} + e\epsilon \right).
$$

$\square$

Next we show that when $\alpha$ is reasonably large, we have $f_i^{[\alpha]}(e_i) - f_j^{[\alpha]}(e_i)$ increasing for all $i \neq j$, which then implies that the loss is decreasing.

**Lemma 7** (Loss Monotonicity). *In the same setting as in Theorem 5, there exists $\alpha_4 = (1 + O(\delta))^{\frac{1}{r-1}} \left( \frac{R_{\max}}{r} \right)^{\frac{1}{r-1}}$ such that the loss is monotonically decreasing for $\alpha \in [\alpha_4, 1]$.*

**Proof of Lemma 7.** To prove that the loss is monotonically decreasing, we only need to show that for any $i \in [k]$ and any $x \in \mathcal{S}_i$, $f_i^{[\alpha]}(x) - f_j^{[\alpha]}(x)$ is monotonically increasing for $j \neq i$.

Same as in Lemma 5, it's easy to prove that for $x \in \mathcal{S}_k$, $f_k(x) - f_j(x)$ with $j \neq k$ monotonically increases for $\alpha \in [0, 1]$. So we focus on other classes.

For $i \neq k$, we show that $\frac{\partial}{\partial \alpha} \left( f_i^{[\alpha]}(x) - f_j^{[\alpha]}(x) \right) > 0$ for $x \in \mathcal{S}_i$ when $\alpha \geq \alpha_4$,

$$
\begin{aligned}
&\frac{\partial}{\partial \alpha} \left( f_i^{[\alpha]}(e_i) - f_j^{[\alpha]}(e_i) \right) \\
&= \frac{\partial}{\partial \alpha} \left( \left[ (1-\alpha) \left\langle W_{i,:}^{(0)}, x \right\rangle + \alpha \left\langle W_{i,:}^{(T)}, x \right\rangle \right]^r - \left[ (1-\alpha) \left\langle W_{j,:}^{(0)}, x \right\rangle + \alpha \left\langle W_{j,:}^{(T)}, x \right\rangle \right]^r + \alpha \left( b_i^{(T)} - b_j^{(T)} \right) \right) \\
&\geq r \left[ (1-\alpha) \left\langle W_{i,:}^{(0)}, x \right\rangle + \alpha \left\langle W_{i,:}^{(T)}, x \right\rangle \right]^{r-1} \left( \left\langle W_{i,:}^{(T)}, x \right\rangle - \left\langle W_{i,:}^{(0)}, x \right\rangle \right) - \left( b_k^{(T)} - b_i^{(T)} \right) - O(\delta^r) \\
&\geq r \alpha^{r-1} \left[ \left\langle W_{i,:}^{(T)}, x \right\rangle \right]^{r-1} \left( \left\langle W_{i,:}^{(T)}, x \right\rangle - \left\langle W_{i,:}^{(0)}, x \right\rangle \right) - \left( b_k^{(T)} - b_i^{(T)} \right) - O(\delta^r) \\
&\geq r \alpha^{r-1} \left( 1 - O \left( \frac{\delta}{\Delta_{\min}^{1/r}} \right) \right) \left[ \left\langle W_{i,:}^{(T)}, x \right\rangle \right]^r - \left( b_k^{(T)} - b_i^{(T)} \right) (1 + O(\delta^r)) \\
&> 0,
\end{aligned}
$$

where the second last inequality uses $\left\langle W_{i,:}^{(0)}, x \right\rangle / \left\langle W_{i,:}^{(T)}, x \right\rangle \leq O \left( \delta / \Delta_{\min}^{1/r} \right)$. The last inequality requires

$$
r \alpha^{r-1} \geq (1 + O(\delta)) \frac{b_k^{(T)} - b_i^{(T)}}{\left[ W_{i,i}^{(T)} \right]^r}
$$

which is satisfied as long as $\alpha \geq (1 + O(\delta))^{\frac{1}{r-1}} \left( \frac{R_{\max}}{r} \right)^{\frac{1}{r-1}}$ where $R_{\max} = \max_{i \in [k-1]} \Delta_i / [W_{i,i}^{(T)}]^r$.. $\square$

### B.3 PLATEAU FOR DEEP FULLY-CONNECTED NETWORKS

In this section, we consider fully-connected neural networks as defined in Section 3 and prove that both the error and loss curves have plateau. We restate Theorem 1 as follows.

**Theorem 1.** *Suppose the network is defined as in Equation* (1) *and suppose the weights satisfy* $\left\|V_i^{(0)}\right\| \le \delta, \left\|V_i^{(T)}\right\| \le V_{\max}$ *for all layers* $i \in [r]$. *On a* $k$-*class balanced dataset whose inputs have* $\ell_2$ *norm at most* 1, *if Assumption 1 holds, for any* $\epsilon > 0$, *as long as* $\delta < \min\left(\frac{\epsilon^{1/r}}{r}, \frac{1}{r^2}, \left(\frac{1}{2e}\right)^{\frac{2}{r-2}}\right)$, *there exist* $\alpha_1 = \frac{\delta}{\Delta_{\min}}, \alpha_2 = \left(\frac{1}{1+\sqrt{\delta}}\right)^{\frac{r}{r-1}} \left(\frac{\Delta_{\min}}{2V_{\max}^r}\right)^{\frac{1}{r-1}}$ *and* $\alpha_3 = \frac{\epsilon^{1/r}}{V_{\max}}$ *such that*

1. *for all* $\alpha \in [\alpha_1, \alpha_2]$, *the error is* $1 - 1/k$;

2. *for all* $\alpha \in [0, \alpha_3]$, *we have* $\log k - 2e\epsilon \le \frac{1}{N} L\left(\left\{V_i^{[\alpha]}\right\}, b^{[\alpha]}\right) \le \log k + \alpha \Delta_{\max} + 2e\epsilon$, *where* $N$ *is the number of training examples.*

**Proof of Theorem 1.** This theorem directly follows from Lemma 8 and Lemma 9. $\qquad \square$

We separately prove the plateau of error interpolation in Lemma 8 and the plateau of loss interpolation in Lemma 9. Then, Theorem 1 is simply a combination of Lemma 8 and Lemma 9. For convenience, we denote $h(x) := V_r \sigma\left(V_{r-1} \cdots \sigma(V_1 x) \cdots\right)$ in the proof.

**Lemma 8.** *In the setting of Theorem 1, there exist* $\alpha_1 = \frac{\delta}{\Delta_{\min}}$ *and* $\alpha_2 = \left(\frac{1}{1+\sqrt{\delta}}\right)^{\frac{r}{r-1}} \left(\frac{\Delta_{\min}}{2V_{\max}^r}\right)^{\frac{1}{r-1}}$ *such that the error is* $1 - 1/k$ *for any interpolation point* $\alpha \in [\alpha_1, \alpha_2]$.

**Proof of Lemma 8.** Recall that the network output under input $x$ is $g(x) := V_r \sigma\left(V_{r-1} \cdots \sigma(V_1 x) \cdots\right) + b$. Similar as in the proof of Lemma 4, we only need to show that for all $\alpha \in [\alpha_1, \alpha_2]$, we have

$$g_i^{[\alpha]}(x) < g_k^{[\alpha]}(x)$$

for all $i \ne k$ and all samples $x$, which immediately implies the error is $1 - 1/k$.

**For** $\alpha \in \left[\alpha_1, \frac{\sqrt{\delta}}{V_{\max}}\right)$. If $\alpha_1 = \frac{\delta}{\Delta_{\min}} \ge \frac{\sqrt{\delta}}{V_{\max}}$, we only need to consider the case when $\alpha \in \left[\frac{\sqrt{\delta}}{V_{\max}}, \alpha_2\right]$. So here we assume $\frac{\delta}{\Delta_{\min}} < \frac{\sqrt{\delta}}{V_{\max}}$. We can lower bound $g_k^{[\alpha]}(x) - g_i^{[\alpha]}(x)$ as

$$
\begin{aligned}
g_k^{[\alpha]}(x) - g_i^{[\alpha]}(x) &= h_k^{[\alpha]}(x) + b_k^{[\alpha]} - h_i^{[\alpha]}(x) - b_i^{[\alpha]} \\
&\ge \alpha \Delta_{\min} - 2 \prod_{j \in [r]} \left\|(1-\alpha)V_j^{(0)} + \alpha V_j^{(T)}\right\| \\
&\ge \alpha \Delta_{\min} - 2\left(\delta + \alpha V_{\max}\right)^r,
\end{aligned}
$$

where the first inequality holds because $b_k^{[\alpha]} - b_i^{[\alpha]} \ge \alpha \Delta_{\min}$ and $\left|h_k^{[\alpha]}(x)\right|, \left|h_i^{[\alpha]}(x)\right| \le \prod_{j \in [r]} \left\|(1-\alpha)V_j^{(0)} + \alpha V_j^{(T)}\right\|$. The second inequality uses $\left\|(1-\alpha)V_j^{(0)} + \alpha V_j^{(T)}\right\| \le (1-\alpha)\left\|V_j^{(0)}\right\| + \alpha\left\|V_j^{(T)}\right\| \le \delta + \alpha V_{\max}$.

Since $\alpha \in \left[\frac{\delta}{\Delta_{\min}}, \frac{\sqrt{\delta}}{V_{\max}}\right)$, we have

$$
\begin{aligned}
g_k^{[\alpha]}(x) - g_i^{[\alpha]}(x) &\ge \frac{\delta}{\Delta_{\min}} \Delta_{\min} - 2\left(\delta + \frac{\sqrt{\delta}}{V_{\max}} V_{\max}\right)^r, \\
&\ge \delta - 2\left(\left(1 + \frac{1}{r}\right)\sqrt{\delta}\right)^r, \\
&\ge \delta - 2e\delta^{r/2} \\
&> 0,
\end{aligned}
$$

where the second inequality assumes $\delta \le 1/r^2$ and the last inequality assumes $\delta < \left(\frac{1}{2e}\right)^{\frac{2}{r-2}}$.

**For** $\alpha \in \left[ \frac{\sqrt{\delta}}{V_{\max}}, \alpha_2 \right]$. Similar as above, we only need to show that $\alpha \Delta_{\min} - 2 \left( \delta + \alpha V_{\max} \right)^r > 0$. Since $\alpha \geq \frac{\sqrt{\delta}}{V_{\max}}$, we have $\delta \leq \sqrt{\delta} \alpha V_{\max}$. Therefore, we have

$$\alpha \Delta_{\min} - 2 \left( \delta + \alpha V_{\max} \right)^r \geq \alpha \Delta_{\min} - 2 \left( \left( 1 + \sqrt{\delta} \right) \alpha V_{\max} \right)^r > 0,$$

where the second inequality holds as long as $\alpha \leq \alpha_2 := \left( \frac{1}{2} \right)^{\frac{1}{r-1}} \left( \frac{1}{1+\sqrt{\delta}} \right)^{\frac{r}{r-1}} \left( \frac{\Delta_{\min}}{V_{\max}^r} \right)^{\frac{1}{r-1}}$. $\qquad \square$

Next, we show that for $\alpha \in [0, \frac{\epsilon^{1/r}}{V_{\max}}]$, the loss cannot decrease by much. Similar as in Lemma 6, we prove that the signal is very small and the logit is dominated by the bias term. This then gives a lower and upper bounds for the loss.

**Lemma 9.** *In the setting of Theorem 1, there exists* $\alpha_3 = \frac{\epsilon^{1/r}}{V_{\max}}$ *such that for all* $\alpha \in [0, \alpha_3]$

$$\log k - 2e\epsilon \leq \frac{1}{N} L \left( \left\{ V_i^{[\alpha]} \right\}, b^{[\alpha]} \right) \leq \log k + \alpha \Delta_{\max} + 2e\epsilon,$$

*where $N$ is the number of samples.*

**Proof of Lemma 9.** Since $\alpha \leq \alpha_3 = \frac{\epsilon^{1/r}}{V_{\max}}$ and $\delta \leq \frac{\epsilon^{1/r}}{r}$, we have

$$\left\| h^{[\alpha]}(x) \right\| \leq \left( \delta + \alpha V_{\max} \right)^r \leq e\epsilon$$

for all input $x$.

Similar as in the proof of Lemma 6, we can show that

$$\log k - 2e\epsilon \leq \frac{1}{N} L \left( \left\{ V_i^{[\alpha]} \right\}, b^{[\alpha]} \right) \leq \log k + \alpha \Delta_{\max} + 2e\epsilon,$$

where we have an additional factor of 2 before $e\epsilon$ because now the signal can be positive or negative. Here $N$ is the number of samples. $\qquad \square$

## C  PROOF OF TRAINING DYNAMICS

In this section, we give the complete proof of Theorem 2.

**Theorem 2.** *Suppose the neural network, dataset and optimization procedure are as defined in Section 4. Suppose initialization scale $\delta \leq \Theta(1)$, noise level $\sigma \leq \widetilde{\Theta}(1)$, dimension $d \geq \widetilde{\Theta}(1/\delta^{2r-2})$ and number of samples $N \geq \widetilde{\Theta}(1/\delta^{r-1})$, with probability at least $0.99$ in the initialization, there exists time $T = \Theta(\log(1/\delta)/\delta^{r-2})$ such that we have*

1. *zero error: for all different $i, j \in [k]$ and for all $x \in \mathcal{S}_i$, $f_i^{(T)}(x) \geq f_j^{(T)}(x) + \Omega(1)$;*

2. *bias gap: $b_{i^*}^{(T)} - \max_{i \neq i^*} b_i^{(T)} \geq \Omega(1)$ with $i^* = \arg\max_{i \in [k]} b_i^{(T)}$.*

**Proof of Theorem 2.** This theorem directly follows from Proposition 1. $\qquad \square$

We consider the $r$-homogeneous-weight network as defined in Section 4. Our simple model simulates a depth-$r$ ReLU/linear network with bias on the output layer, in the sense that the weights signal is $r$-homogeneous while the bias is $1$-homogeneous in the parameters.

Next, we prove Proposition 1 while leaving the proof of supporting lemmas into Section C.1. Through the proof of Proposition 1, we restate the lemmas when we use it for the convenience of readers.

**Proposition 1** (Induction Hypothesis)**.** *In the same setting of Theorem 2, with probability at least $0.99$ in initialization, there exist time points $0 =: s_1 < t_1 < s_2 < t_2 < \cdots < s_{k-1} < t_{k-1} < s_k := T$ with $t_i - s_i = \Theta(\log(1/\delta)/\delta^{r-2})$ and $s_{i+1} - t_i = \Theta(1)$ for $i \in [k-1]$ such that for any $t \in [s_i, s_{i+1}]$,*

1. *(**classes not yet learned**) for any class $j, j' \geq i+1$, we have (1) $b_j^{(t)} \geq \max_{i' \in [k]} b_{i'}^{(t)} - O(\delta^r)$, (2) $\left| b_j^{(t)} - b_{j'}^{(t)} \right| \leq O(\delta^r)$ and (3) $W_{j,j}^{(t)} \leq O(\delta)$;*

2. *(classes already learned) for any class $j \leq i - 1$, we have (1) $b_j^{(t)} \leq \max_{i' \in [k]} b_{i'}^{(t)} - \Omega(1)$, (2) $f_j^{(t)}(x) \geq f_{i'}^{(t)}(x) + \Omega(1)$ for $i' \neq j, x \in S_j$ and (3) $W_{j,j}^{(t)} \geq \Omega(1)$;*

3. *(parameters movement) (1) for any $j \in [k], \Theta(\delta) = W_{j,j}^{(0)} < W_{j,j}^{(t)}$, (2) for any distinct $j, j' \in [k], 0 < W_{j,j'}^{(t)} \leq O(\delta)$ and (3) for any $j, j' \in [k]$ and any $x \in S_{j'}, \left| \left\langle W_{j,:}^{(t)}, \xi_x \right\rangle \right| \leq \min \left( O(\delta), W_{j,j'}^{(t)} \right)$.*

**Proof of Proposition 1.** Through the proof, we assume the conditions in Theorem 2 hold in all the lemmas without explicitly stated. At the initialization, we have the following properties with probability at least 0.99.

**Lemma 10** (Initialization). *With probability at least 0.99 in the initialization, we have*

1. *for all $j, j' \in [k], W_{j,j'}^{(0)} = \Theta(\delta)$;*

2. *for all distinct $j, j' \in [k], \left| W_{j,j}^{(0)} - W_{j',j'}^{(0)} \right| = \Theta(\delta)$;*

3. *for all $x \in S, \|\xi_x\| \leq O(\sigma)$;*

4. *for all distinct $x, x' \in S, \left| \left\langle \bar{\xi}_x, \bar{\xi}_{x'} \right\rangle \right| \leq O \left( \frac{\sqrt{\log(N)}}{\sqrt{d}} \right)$.*

5. *for all $j \in [k]$ and all $x \in S, \left| \left\langle \bar{\xi}_x, e_j \right\rangle \right|, \left| \left\langle \bar{\xi}_x, \bar{W}_{j,:}^{(0)} \right\rangle \right| \leq O \left( \frac{\sqrt{\log(N)}}{\sqrt{d}} \right)$.*

*Without loss of generality, we assume $W_{1,1}^{(0)} > W_{2,2}^{(0)} > \cdots > W_{k,k}^{(0)}$.*

It's not hard to verify that the induction hypothesis holds at the initialization [4]. For any $i \in [k-1]$, assuming the induction hypothesis holds for time $[0, s_i]$, now we prove that it continues to hold in $[s_i, s_{i+1}]$. Next, we first prove the first two properties in the Proposition 1 and leave the last one at the end.

The learning of $x \in S_i$ can be divided into four stages:

1. **Stage 0** for $t \in [s_i, t_i]$ with $t_i - s_i = O(\log(1/\delta)/\delta^{r-2})$. During this stage, $W_{i,i}^{(t)}$ grows to a small constant $\mu_0$.

2. **Stage 1** for $t \in [t_i, t_i^{(w)}]$ with $t_i^{(w)} - t_i = O(1)$. In this stage, $W_{i,i}^{(t)}$ grows from $\mu_0$ to a large constant $\mu_1$.

3. **Stage 2** for $t \in [t_i^{(w)}, t_i^{(u)}]$ with $t_i^{(u)} - t_i^{(w)} = O(1)$. At the end of this stage, we have $\min_{x \in S_i} u_i^{(t)}(x) \geq 1 - \mu_2$ for a small constant $\mu_2$.

4. **Stage 3** for $t \in [t_i^{(u)}, t_i^{(b)}]$ with $t_i^{(b)} - t_i^{(u)} = O(1)$, where $t_i^{(b)} = s_{i+1}$. During this stage, we have $b_i^{(t)} - b_k^{(t)}$ decreases to $-\mu_3$ with $\mu_3$ a positive constant.

Next, we consider these four stages one by one.

**Stage 0.**   We show that $W_{i,i}^{(t)}$ increases faster than $W_{i+1,i+1}^{(t)}$ so that $W_{i,i}^{(t)}$ reaches a constant while $W_{i+1,i+1}^{(t)}$ is still $O(\delta)$. We use the following lemma to characterize the increasing rate of $W_{i+1,i+1}^{(t)}$ and $W_{i,i}^{(t)}$.

---

[4]We will maintain a stronger bound on $\left| \left\langle W_{j,:}^{(t)}, \xi_x \right\rangle \right|$ by proving $\left| \left\langle W_{j,:}^{(t)}, \xi_x \right\rangle \right| \leq O(\sqrt{\log N} \sigma \delta)$, which implies $\left| \left\langle W_{j,:}^{(t)}, \xi_x \right\rangle \right| \leq O(\delta)$ as long as $\sigma \leq O(1/\sqrt{\log N})$.

**Lemma 11.** *For any $j \in [k]$, we have*

$$\frac{\partial W_{j,j}^{(t)}}{\partial t} \geq -O\left(\frac{\delta^{r-1}\sqrt{\log N}\sigma}{\sqrt{d}}\right).$$

*If $\min_{x \in S_j}(1 - u_j(x)) \geq \Omega(1)$, we further have*

$$\left(1 - O(\sqrt{\log N}\sigma)\right)\frac{k}{N}\sum_{x \in S_j}(1 - u_j(x))rW_{j,j}^{r-1} \leq \frac{\partial W_{j,j}^{(t)}}{\partial t} \leq \left(1 + O(\sqrt{\log N}\sigma)\right)\frac{k}{N}\sum_{x \in S_j}(1 - u_j(x))rW_{j,j}^{r-1}.$$

It's not hard to verify that $\min_{x \in S_i}\left(1 - u_i^{(t)}(x)\right), \min_{x \in S_{i+1}}\left(1 - u_{i+1}^{(t)}(x)\right) \geq \Omega(1)$, so we have

$$\frac{\partial W_{i,i}^{(t)}}{\partial t} \geq \left(1 - O(\sqrt{\log N}\sigma)\right)\frac{k}{N}\sum_{x \in S_i}\left(1 - u_i^{(t)}(x)\right)r\left[W_{i,i}^{(t)}\right]^{r-1},$$

$$\frac{\partial W_{i+1,i+1}^{(t)}}{\partial t} \leq \left(1 + O(\sqrt{\log N}\sigma)\right)\frac{k}{N}\sum_{x \in S_{i+1}}\left(1 - u_{i+1}^{(t)}(x)\right)r\left[W_{i+1,i+1}^{(t)}\right]^{r-1}.$$

We can upper bound $1 - u_{i+1}^{(t)}(x)$ for any $x \in S_{i+1}$ as follows,

$$1 - u_{i+1}^{(t)}(x) = \frac{\sum_{i' \in [k]}\exp\left(f_{i'}^{(t)}(x)\right) - \exp\left(f_{i+1}^{(t)}(x)\right)}{\sum_{i' \in [k]}\exp\left(f_{i'}^{(t)}(x)\right)}$$

$$\leq \frac{\sum_{i' \in [k]}\exp\left(b_{i'}^{(t)}\right) - \exp\left(b_{i+1}^{(t)}\right)}{\sum_{i' \in [k]}\exp\left(b_{i'}^{(t)}\right)}(1 + O(\delta^r)),$$

where the inequality uses $\left|\left\langle W_{i',:}^{(t)}, x\right\rangle\right| \leq O(\delta)$ for every $i' \in [k]$.

We can lower bound $1 - u_i^{(t)}(x)$ for $x \in S_i$ as follows,

$$1 - u_i^{(t)}(x) = \frac{\sum_{i' \in [k]}\exp\left(f_{i'}^{(t)}(x)\right) - \exp\left(f_i^{(t)}(x)\right)}{\sum_{i' \in [k]}\exp\left(f_{i'}^{(t)}(x)\right)}$$

$$\geq \frac{\sum_{i' \in [k]}\exp\left(b_{i'}^{(t)}\right) - \exp\left(b_i^{(t)}\right)}{\sum_{i' \in [k]}\exp\left(b_{i'}^{(t)}\right)}(1 - O(\mu_0^r))$$

$$\geq \frac{\sum_{i' \in [k]}\exp\left(b_{i'}^{(t)}\right) - \exp\left(b_{i+1}^{(t)}\right)}{\sum_{i' \in [k]}\exp\left(b_{i'}^{(t)}\right)}(1 - O(\mu_0^r) - O(\delta^r)).$$

The first inequality uses $\left|\left\langle W_{i',:}^{(t)}, x\right\rangle\right| \leq O(\mu_0)$ for every $i' \in [k]$. The second inequality uses $b_i^{(t)} \leq b_{i+1}^{(t)} + O(\delta^r)$, which is guaranteed by the following lemma.

**Lemma 2** (Bias Gap Control I). *For any different $j', j \in [k]$, if $W_{j',j'} \geq W_{j,j}, W_{j,j} \leq O(\delta)$ and $b_{j'} - b_j \geq O(\delta^r), b_j \geq \max_{i' \in [k]}b_{i'} - O(\delta^r)$, we have $\dot{b}_{j'} - \dot{b}_j < 0$.*

According to Lemma 10, we know there exists constant $C > 1$ such that $W_{i,i}^{(0)} \geq CW_{i+1,i+1}^{(0)}$. Choose constant $S$ such that $S^{\frac{1}{r-2}} = \sqrt{C}$ and $W_{i,i}^{(0)} \geq S^{\frac{1}{r-2}}\sqrt{C}W_{i+1,i+1}^{(0)}$. Choosing $\mu_0$ as small constants and $\sigma \leq O(1/\sqrt{\log N}), \delta \leq O(1)$, we have

$$\left(1 + O(\sqrt{\log N}\sigma)\right)\max_{x \in S_{i+1}}\left(1 - u_{i+1}^{(t)}(x)\right) \leq S\left(1 - O(\sqrt{\log N}\sigma)\right)\min_{x \in S_i}\left(1 - u_i^{(t)}(x)\right).$$

We can also lower bound $1 - u_i^{(t)}(x)$ for $x \in \mathcal{S}_i$ by a constant,

$$
\begin{aligned}
1 - u_i^{(t)}(x) &\geq \frac{\sum_{i' \in [k]} \exp\left(b_{i'}^{(t)}\right) - \exp\left(b_i^{(t)}\right)}{\sum_{i' \in [k]} \exp\left(b_{i'}^{(t)}\right)} (1 - O(\mu_0^r)) \\
&\geq \frac{\exp\left(b_{i+1}^{(t)}\right)}{\sum_{i' \in [k]} \exp\left(b_{i'}^{(t)}\right)} (1 - O(\mu_0^r)) \\
&\geq \Omega(1),
\end{aligned}
$$

where the last inequality holds because $\mu_0$ is a small constant and $b_{i+1}^{(t)} \geq \max_{i' \in [k]} b_{i'}^{(t)} - O(\delta^r)$.

**Lemma 12** (Adapted from Lemma C.19 in Allen-Zhu & Li (2020)). *Let $r \geq 3$ be a constant and let $\{W_{i,i}^{(t)}, W_{j,j}^{(t)}\}_{t \geq 0}$ be two positive sequences updated as*

$$
\frac{\partial W_{i,i}^{(t)}}{\partial t} \geq C_t \left[W_{i,i}^{(t)}\right]^{r-1} \text{ for some } C_t = \Theta(1),
$$

$$
\frac{\partial W_{j,j}^{(t)}}{\partial t} \leq S C_t \left[W_{j,j}^{(t)}\right]^{r-1} \text{ for some } S = \Theta(1).
$$

*Suppose $W_{i,i}^{(0)} \geq W_{j,j}^{(0)} S^{\frac{1}{r-2}} (1 + \Omega(1))$, then we must have for every $A = O(1)$, let $t_i$ be the first time such that $W_{i,i}^{(t_i)} \geq A$, then*

$$
W_{j,j}^{(t_i)} \leq O(W_{j,j}^{(0)}).
$$

Then, according to Lemma 12, we know that there exists $t_i = O(\log(1/\delta)/\delta^{r-2})$ such that $W_{i,i}^{(t_i)} = \mu_0$ and $W_{i+1,i+1}^{(t_i)} \leq O(\delta)$. By similar argument, we also know $W_{j,j}^{(t_i)} \leq O(\delta)$ for any $j \geq i + 1$.

**Stage 1.** In this stage, we show that $W_{i,i}^{(t)}$ grows to a large constant $\mu_1$ within constant time. Since $W_{i,i}^{(t)} \leq \mu_1$ and $b_{i,i}^{(t)} \leq b_{i+1,i+1}^{(t)} + O(\delta^r)$, we have

$$
1 - u_i^{(t)}(x) \geq \Omega(1),
$$

for all $x \in \mathcal{S}_i$. This further implies,

$$
\begin{aligned}
\frac{\partial W_{i,i}^{(t)}}{\partial t} &\geq \left(1 - O(\sqrt{\log N}\sigma)\right) \frac{k}{N} \sum_{x \in \mathcal{S}_i} \left(1 - u_i^{(t)}(x)\right) r \left[W_{i,i}^{(t)}\right]^{r-1} \\
&\geq \Omega(1),
\end{aligned}
$$

where the inequality also uses $W_{i,i}^{(t)} \geq \mu_0$. Since the increasing rate is at least a constant, we know $W_{i,i}^{(t)}$ grows to $\mu_1$ in constant time. For any $j \geq i + 1$, since the increasing rate of $W_{j,j}^{(t)}$ is merely $O(\delta^{r-1})$, we know $W_{j,j}^{(t)}$ remains as $O(\delta)$ through Stage 1.

**Stage 2.** In this stage, we prove that $u_i^{(t)}(x)$ for any $x \in \mathcal{S}_i$ grows to $1 - \mu_2$ with $\mu_2$ a small constant. We use the following lemma to characterize the increasing rate of $f_i^{(t)}(x) - f_j^{(t)}(x)$.

**Lemma 13.** *For any $x \in \mathcal{S}_i$ and any $j \neq i$, if $1 - u_i(x) \geq \Omega(1)$, we have*

$$
\frac{\partial}{\partial t} (f_i(x) - f_j(x)) \geq \Omega(W_{i,i}^{2r-2}) - O(1).
$$

Since $u_i^{(t)}(x) \leq 1 - \mu_2$, we know $1 - u_i^{(t)}(x) \geq \Omega(1)$. For any $j \neq i$, we have

$$
\begin{aligned}
\frac{\partial}{\partial t} \left(f_i^{(t)}(x) - f_j^{(t)}(x)\right) &\geq \Omega\left(\left[W_{i,i}^{(t)}\right]^{2r-2}\right) - O(1) \\
&\geq \Omega(1),
\end{aligned}
$$

where the last inequality holds because $W_{i,i}^{(t)} \geq \mu_1$ with $\mu_1$ a large enough constant.

The next lemma guarantees that at the beginning of Stage 2, we have $b_i^{(t)} - b_j^{(t)} \geq -O(1)$, which then implies $f_i^{(t)}(x) - f_j^{(t)}(x) \geq -O(1)$.

**Lemma 14** (Bias Gap Control III). *For any different $i, j \in [k]$, if $W_{i,i} \leq O(1), W_{j,j} \leq O(\delta)$ and $b_i - b_j \leq -O(1)$, we have*

$$\dot{b}_i - \dot{b}_j > 0.$$

Let $C$ be a constant such that $f_i^{(t)}(x) - f_j^{(t)}(x) \geq C$ for every $j \neq i$ implies $u_i(x) \geq 1 - \mu_2$. Since at the at the beginning of Stage 2, we have $f_i^{(t)}(x) - f_j^{(t)}(x) \geq -O(1)$, within constant time, we have $f_i^{(t)}(x) - f_j^{(t)}(x) \geq C$ for every $j \neq i$ and $u_i(x) \geq 1 - \mu_2$.

**Lemma 15** (Accuracy Monotonicity). *Given any positive constant $C_2$, there exists positive constant $C_1$ such that for all different $i, j \in [k]$, as long as $W_{i,i} \geq C_1$ and $f_i(x) - f_j(x) \leq C_2$ for any $x \in \mathcal{S}_i$, we have $\frac{\partial(f_i(x) - f_j(x))}{\partial t} > 0$.*

According to Lemma 15, by choosing large enough $\mu_1$, we can ensure that $f_i^{(t)}(e_i) - f_j^{(t)}(e_i) \geq C$ and $u_i(e_i) \geq 1 - \mu_2$ throughout the training. Note that once $W_{i,i}^{(t)}$ rises to $\mu_1$, it will stay at least $\mu_1 - O(\delta)$ throughout the training, according to the gradient lower bound in Lemma 11.

**Stage 3.** In this stage, we prove that within constant time we have $b_i^{(t)} - b_k^{(t)} \leq -\mu_3$. The following lemma shows that $b_i^{(t)} - b_k^{(t)}$ decreases in at least a constant rate.

**Lemma 3** (Bias Gap Control II). *There exist small positive constants $C_1, C_2$ such that for any $j \in [k-1]$ and any $x \in \mathcal{S}_j$, if $1 - u_j(x) \leq C_1, W_{k,k} \leq O(\delta)$ and $b_j - b_k \geq -C_2$, we have $\dot{b}_j - \dot{b}_k < -\Omega(1)$.*

Choosing $\mu_2 = C_1, \mu_3 = C_2$ where $C_1, C_2$ are from Lemma 3, so we know that $b_i^{(t)} - b_k^{(t)}$ decreases at a constant rate until $b_i^{(t)} - b_k^{(t)} \leq -\mu_3$. At time $t_i^{(u)}$, we know that $b_i^{t_i^{(u)}} - b_k^{t_i^{(u)}} \leq O(\delta^r)$. So within constant time, we have $b_i^{s_{i+1}} - b_k^{s_{i+1}} = -\mu_3$. By Lemma 3, we also know that for any $t \geq s_{i+1}$, we have $b_i^{(t)} - b_k^{(t)} \leq -\mu_3$.

The following lemma shows that $b_k^{(t)}$ is close to the maximum bias.

**Lemma 1** (Coupling Biases). *Assuming $W_{j',j'}, W_{j,j} \leq O(\delta)$ and $b_{j'}, b_j \geq \max_{i' \in [k]} b_{i'} - O(\delta^r)$, we have $\dot{b}_{j'} - \dot{b}_j > 0$ if $b_{j'} - b_j \leq -\mu\delta^r$, and $\dot{b}_{j'} - \dot{b}_j < 0$ if $b_{j'} - b_j \geq +\mu\delta^r$ for some positive constant $\mu$.*

Combining Lemma 1 and Lemma 2, we know that throughout the training $b_k^{(t)} \geq \max_{i' \in [k]} b_{i'}^{(t)} - O(\delta^r)$. Therefore, we have $b_i^{(t)} - \max_{i' \in [k]} b_{i'}^{(t)} \leq -\Omega(1)$ for $t \geq s_{i+1}$.

Finally, let's bound the movement of different parameters.

**Monotonicity of diagonal terms:** For $j \in [k-1]$, according to Lemma 11 we know $W_{j,j}^{(t)}$ can only start decreasing when it exceeds a large constant and can only decrease by at most $O(\delta)$ through the algorithm. By choosing $\delta \leq O(1)$, we can ensure that $W_{j,j}^{(0)} < W_{j,j}^{(t)}$ for any $t$. For $W_{k,k}^{(t)}$, we know it monotonically increases since we always have $1 - u_k^{(t)}(x) \geq \Omega(1)$ for $x \in \mathcal{S}_k$. This is because $W_{k,k}^{(t)}$ remains as small as $O(\delta)$ through the algorithm and $b_k^{(t)} - b_{k-1}^{(t)} \leq O(1)$.

**Bounding non-diagonal terms:** We use the following lemma to prove that $\widetilde{\Omega}(\delta) < W_{j,j'}^{(t)} \leq O(\delta)$ for $j \neq j'$.

**Lemma 16.** *For any $j \neq j'$, we have $T\dot{W}_{j,j'} \leq O(\delta)$. Furthermore, there exists absolute constant $\mu > 0$ such that if $0 < W_{j,j'} < \frac{\mu\delta}{\log^{\frac{1}{r-2}}(1/\delta)}$, we have $T\dot{W}_{j,j'} \geq -\frac{\mu\delta}{2\log^{\frac{1}{r-2}}(1/\delta)}$.*

The first property in Lemma 16 guarantees that the increasing rate is so small that that the total increase within $T$ time is only $O(\delta)$, which then implies that $W_{j,j'}^{(t)} \leq O(\delta)$ through the training. The second property in Lemma 16 guarantees that once $W_{j,j'}^{(t)}$ falls below $\frac{\mu\delta}{\log^{\frac{1}{r-2}}(1/\delta)}$, its decreasing rate is so small that that the total decrease within $T$ time is only $\frac{\mu\delta}{2\log^{\frac{1}{r-2}}(1/\delta)}$, which then implies that $W_{j,j'}^{(t)} > \widetilde{\Omega}(\delta)$ through the training.

**Bounding noise correlations:** The following lemma shows that the total change of $\left\langle W_{j,:}^{(t)}, \xi_x \right\rangle$ within $T$ time is only $O(\sqrt{\log N}\sigma\delta)$. Since at initialization, we know $\left|\left\langle W_{j,:}^{(0)}, \xi_x \right\rangle\right| \leq O(\sqrt{\log N}\sigma\delta)$, we conclude that $\left|\left\langle W_{j,:}^{(t)}, \xi_x \right\rangle\right| \leq O(\sqrt{\log N}\sigma\delta)$ throughout the training. Since $W_{j,j'}^{(t)} \geq \widetilde{\Omega}(\delta)$, as long as $\sigma \leq \widetilde{O}(1)$, we also have $\left|\left\langle W_{j,:}^{(t)}, \xi_x \right\rangle\right| \leq W_{j,j'}$ for $x \in \mathcal{S}_{j'}$.

**Lemma 17.** *For every $j \in [k]$ and every $x \in \mathcal{S}$, we have*

$$\left|\left\langle \dot{W}_{j,:}, \xi_x \right\rangle\right| \cdot T \leq O\left(\sqrt{\log N}\sigma\delta\right)$$

$\square$

## C.1 PROOF OF LEMMAS

**Lemma 10** (Initialization). *With probability at least $0.99$ in the initialization, we have*

1. *for all $j, j' \in [k]$, $W_{j,j'}^{(0)} = \Theta(\delta)$;*

2. *for all distinct $j, j' \in [k]$, $\left|W_{j,j}^{(0)} - W_{j',j'}^{(0)}\right| = \Theta(\delta)$;*

3. *for all $x \in \mathcal{S}$, $\|\xi_x\| \leq O(\sigma)$;*

4. *for all distinct $x, x' \in \mathcal{S}$, $\left|\left\langle \bar{\xi}_x, \bar{\xi}_{x'} \right\rangle\right| \leq O\left(\frac{\sqrt{\log(N)}}{\sqrt{d}}\right)$.*

5. *for all $j \in [k]$ and all $x \in \mathcal{S}$, $\left|\left\langle \bar{\xi}_x, e_j \right\rangle\right|, \left|\left\langle \bar{\xi}_x, \bar{W}_{j,:}^{(0)} \right\rangle\right| \leq O\left(\frac{\sqrt{\log(N)}}{\sqrt{d}}\right)$.*

*Without loss of generality, we assume $W_{1,1}^{(0)} > W_{2,2}^{(0)} > \cdots > W_{k,k}^{(0)}$.*

**Proof of Lemma 10.** Recall that each $W_{j,j'}^{(0)}$ is independently sampled from $\mathcal{N}(0, \delta^2)$ before taking the absolute value. By standard Gaussian concentration inequality, we know for any $j, j' \in [k]$, with probability at least $1 - \frac{1}{1000k^2}$,

$$W_{j,j'}^{(0)} \leq O(\delta).$$

By anti-concentration inequality of Gaussian polynomials, we know for any $j, j' \in [k]$, with probability at least $1 - \frac{1}{1000k^2}$,

$$W_{j,j'}^{(0)} \geq \Omega(\delta).$$

Also by anti-concentration inequality of Gaussian polynomials, we know for any distinct $j, j' \in [k]$, with probability at least $1 - \frac{1}{1000k^2}$,

$$\left|\left[W_{j,j}^{(0)}\right]^2 - \left[W_{j',j'}^{(0)}\right]^2\right| \geq \Omega(\delta^2),$$

which implies $\left|W_{j,j}^{(0)} - W_{j',j'}^{(0)}\right| \geq \Omega(\delta)$ assuming $W_{j,j}^{(0)}, W_{j',j'}^{(0)} = \Theta(\delta)$.

By the norm concentration of random vectors with independent Gaussian entries, for each $x \in \mathcal{S}$, we have with probability at least $1 - \frac{1}{1000N^2}$,

$$\|\xi_x\| \leq O(\sigma)$$

as long as $d \geq O(\log N)$.

By the concentration of standard Gaussian variable, for any distinct $x, x' \in \mathcal{S}$, we have with probability at least $1 - \frac{1}{1000N^2}$,

$$\left|\langle \bar{\xi}_x, \bar{\xi}_{x'} \rangle\right| \leq O\left(\frac{\sqrt{\log N}}{\sqrt{d}}\right).$$

Similarly, for any $x$ and any $e_j$, we have with probability at least $1 - \frac{1}{1000kN}$,

$$\left|\langle \bar{\xi}_x, e_j \rangle\right| \leq O\left(\frac{\sqrt{\log N}}{\sqrt{d}}\right);$$

for any $x$ and any $\bar{W}_{j,:}^{(0)}$, we have with probability at least $1 - \frac{1}{1000kN}$,

$$\left|\left\langle \bar{\xi}_x, \bar{W}_{j,:}^{(0)} \right\rangle\right| \leq O\left(\frac{\sqrt{\log N}}{\sqrt{d}}\right);$$

Taking a union bound over all these events, we know with probability at least $0.99$ in the initialization, we have

1. for all $j, j' \in [k]$, $W_{i,j}^{(0)} = \Theta(\delta)$;

2. for all distinct $j, j' \in [k]$, $\left|W_{j,j}^{(0)} - W_{j',j'}^{(0)}\right| = \Theta(\delta)$;

3. for all $x \in \mathcal{S}$, $\|\xi_x\| \leq O(\sigma)$;

4. for all distinct $x, x' \in \mathcal{S}$, $\left|\langle \bar{\xi}_x, \bar{\xi}_{x'} \rangle\right| \leq O\left(\frac{\sqrt{\log(N)}}{\sqrt{d}}\right).$

5. for all $j \in [k]$ and all $x \in \mathcal{S}$, $\left|\langle \bar{\xi}_x, e_j \rangle\right|, \left|\left\langle \bar{\xi}_x, \bar{W}_{j,:}^{(0)} \right\rangle\right| \leq O\left(\frac{\sqrt{\log(N)}}{\sqrt{d}}\right).$

$\square$

**Lemma 12** (Adapted from Lemma C.19 in Allen-Zhu & Li (2020)). *Let $r \geq 3$ be a constant and let $\{W_{i,i}^{(t)}, W_{j,j}^{(t)}\}_{t \geq 0}$ be two positive sequences updated as*

$$\frac{\partial W_{i,i}^{(t)}}{\partial t} \geq C_t \left[W_{i,i}^{(t)}\right]^{r-1} \text{ for some } C_t = \Theta(1),$$

$$\frac{\partial W_{j,j}^{(t)}}{\partial t} \leq SC_t \left[W_{j,j}^{(t)}\right]^{r-1} \text{ for some } S = \Theta(1).$$

*Suppose $W_{i,i}^{(0)} \geq W_{j,j}^{(0)} S^{\frac{1}{r-2}} (1 + \Omega(1))$, then we must have for every $A = O(1)$, let $t_i$ be the first time such that $W_{i,i}^{(t_i)} \geq A$, then*

$$W_{j,j}^{(t_i)} \leq O(W_{j,j}^{(0)}).$$

**Proof of Lemma 12.** This lemma directly follows from Lemma C.19 in Allen-Zhu & Li (2020) by taking the continuous time limit and setting $k$ as a constant. $\square$

**Lemma 1** (Coupling Biases). *Assuming $W_{j',j'}, W_{j,j} \leq O(\delta)$ and $b_{j'}, b_j \geq \max_{i' \in [k]} b_{i'} - O(\delta^r)$, we have $\dot{b}_{j'} - \dot{b}_j > 0$ if $b_{j'} - b_j \leq -\mu\delta^r$, and $\dot{b}_{j'} - \dot{b}_j < 0$ if $b_{j'} - b_j \geq +\mu\delta^r$ for some positive constant $\mu$.*

**Proof of Lemma 1.** Let's first write down the time derivative on $b_{j'}$,

$$\dot{b}_{j'} = 1 - \frac{k}{N} \sum_{x \in \mathcal{S}} u_{j'}(x)$$

$$= 1 - \frac{k}{N} \sum_{x \in \mathcal{S}} \frac{\exp\left(\langle W_{j',:}, x \rangle^r + b_{j'}\right)}{\sum_{i' \in [k]} \exp\left(\langle W_{i',:}, x \rangle^r + b_{i'}\right)}$$

For any $x \in \mathcal{S}$, we can bound $\frac{\exp\left(\langle W_{j',:}, x \rangle^r + b_{j'}\right)}{\sum_{i' \in [k]} \exp\left(\langle W_{i',:}, x \rangle^r + b_{i'}\right)}$ as follows,

$$\left| \frac{\exp\left(\langle W_{j',:}, x \rangle^r + b_{j'}\right)}{\sum_{i' \in [k]} \exp\left(\langle W_{i',:}, x \rangle^r + b_{i'}\right)} - \frac{\exp\left(b_{j'}\right)}{\sum_{i' \in [k]} \exp\left(\langle W_{i',:}, x \rangle^r + b_{i'}\right)} \right| \leq O(\delta^r),$$

where we uses $|\langle W_{j',:}, x \rangle| \leq O(\delta) + O(\sqrt{\log N} \sigma \delta) \leq O(\delta)$ assuming $\sigma \leq 1/\sqrt{\log N}$. The similar bound also holds for $\frac{\exp\left(\langle W_{j,:}, x \rangle^r + b_j\right)}{\sum_{i' \in [k]} \exp\left(\langle W_{i',:}, x \rangle^r + b_{i'}\right)}$

If $b_{j'} - b_j \geq \mu \delta^r$, we can now upper bound $\dot{b}_{j'} - \dot{b}_j$ as follows,

$$\dot{b}_{j'} - \dot{b}_j \leq \frac{k}{N} \sum_{x \in \mathcal{S}} \frac{\exp\left(b_j\right) - \exp\left(b_{j'}\right)}{\sum_{i' \in [k]} \exp\left(\langle W_{i',:}, x \rangle^r + b_{i'}\right)} + O(\delta^r)$$

$$\leq \frac{k}{N} \sum_{x \in \mathcal{S}_j \cup \mathcal{S}_{j'}} \frac{\exp\left(b_j\right) - \exp\left(b_{j'}\right)}{\sum_{i' \in [k]} \exp\left(\langle W_{i',:}, x \rangle^r + b_{i'}\right)} + O(\delta^r)$$

$$\leq -\Omega(\mu \delta^r) \cdot \frac{k}{N} \sum_{x \in \mathcal{S}_j \cup \mathcal{S}_{j'}} \frac{\exp\left(b_j\right)}{\sum_{i' \in [k]} \exp\left(\langle W_{i',:}, x \rangle^r + b_{i'}\right)} + O(\delta^r)$$

When $x \in \mathcal{S}_j \cup \mathcal{S}_{j'}$, we can lower bound $\frac{\exp(b_j)}{\sum_{i' \in [k]} \exp\left(\langle W_{i',:}, x \rangle^r + b_{i'}\right)}$ as follows,

$$\frac{\exp\left(b_j\right)}{\sum_{i' \in [k]} \exp\left(\langle W_{i',:}, x \rangle^r + b_{i'}\right)} = \frac{\exp\left(b_j\right)}{\sum_{i' \in [k]} \exp\left(b_{i'}\right) \exp\left(\langle W_{i',:}, x \rangle^r\right)}$$

$$\geq \frac{\exp\left(b_j\right)}{\sum_{i' \in [k]} \exp\left(b_{i'}\right)} \cdot \frac{1}{1 + O(\delta^r)}$$

$$\geq \Omega(1),$$

where the first inequality uses $|\langle W_{i',:}, x \rangle| \leq \delta$ and the second inequality assumes $b_j \geq \max_{i' \in [k]} b_{i'} - O(\delta^r)$ and $\delta$ is at most some small constant.

Therefore, if $b_{j'} - b_j \geq \mu \delta^r$, we have

$$\dot{b}_{j'} - \dot{b}_j \leq -\Omega(\mu \delta^r) + O(\delta^r) < 0,$$

where the second inequality chooses $\mu$ as a large enough constant. Similarly, we can prove that if $b_{j'} - b_j \leq -\mu \delta^r$, we have

$$\dot{b}_{j'} - \dot{b}_j \geq \Omega(\mu \delta^r) - O(\delta^r) > 0.$$

$\square$

**Lemma 2** (Bias Gap Control I). *For any different $j', j \in [k]$, if $W_{j',j'} \geq W_{j,j}, W_{j,j} \leq O(\delta)$ and $b_{j'} - b_j \geq O(\delta^r), b_j \geq \max_{i' \in [k]} b_{i'} - O(\delta^r)$, we have $\dot{b}_{j'} - \dot{b}_j < 0$.*

**Proof of Lemma 2.** We can write down $\dot{b}_{j'} - \dot{b}_j$ as follows,

$$\dot{b}_{j'} - \dot{b}_j = \left(1 - \frac{k}{N} \sum_{x \in \mathcal{S}} u_{j'}(x)\right) - \left(1 - \frac{k}{N} \sum_{x \in \mathcal{S}} u_j(x)\right)$$

$$= \frac{k}{N} \sum_{x \in \mathcal{S}_{j'}} \left(u_j(x) - u_{j'}(x)\right) + \frac{k}{N} \sum_{x \in \mathcal{S} \setminus \mathcal{S}_{j'}} \left(u_j(x) - u_{j'}(x)\right).$$

We first prove that for any $x \in \mathcal{S}_{j'}$, we have $u_j(x) - u_{j'}(x) \leq 0$. We can upper bound $f_j(x)$ and lower bound $f_{j'}(x)$ as follows,

$$f_j(x) = \langle W_{j,:}, x \rangle^r + b_j \leq O(\delta^r) + b_j$$
$$f_{j'}(x) = \langle W_{j',:}, x \rangle^r + b_{j'} \geq b_j.$$

The bound on $f_j(x)$ holds because $\langle W_{j,:}, x \rangle = W_{j,j'} + \langle W_{j,:}, \xi_x \rangle \leq O(\delta) + O(\sqrt{\log N} \sigma \delta) \leq O(\delta)$. The bound on $f_{j'}(x)$ holds because $\langle W_{j',:}, x \rangle = W_{j',j'} + \langle W_{j',:}, \xi_x \rangle \geq \Omega(\delta) - O(\sqrt{\log N} \sigma \delta) > 0$. With the above two bounds, we know that $u_j(x) - u_{j'}(x) \leq 0$ as long as $b_{j'} - b_j \geq O(\delta^r)$.

Same as in the proof of Lemma 1, for each $x \in \mathcal{S} \setminus \mathcal{S}_{j'}$, we can bound $u_{j'}(x), u_j(x)$ as follows,

$$\frac{\exp(b_{j'})}{\sum_{i' \in [k]} \exp(f_{i'}(x))} - O(\delta^r) \leq u_{j'}(x) \leq \frac{\exp(b_{j'})}{\sum_{i' \in [k]} \exp(f_{i'}(x))} + O(\delta^r),$$

$$\frac{\exp(b_j)}{\sum_{i' \in [k]} \exp(f_{i'}(x))} - O(\delta^r) \leq u_j(x) \leq \frac{\exp(b_j)}{\sum_{i' \in [k]} \exp(f_{i'}(x))} + O(\delta^r).$$

Therefore, if $b_{j'} - b_j \geq \mu \delta^r$, we can further upper bound $\dot{b}_{j'} - \dot{b}_j$ as follows,

$$\dot{b}_{j'} - \dot{b}_j \leq \frac{k}{N} \sum_{x \in \mathcal{S} \setminus \mathcal{S}_{j'}} (u_j(x) - u_{j'}(x)).$$

$$\leq \frac{k}{N} \sum_{x \in \mathcal{S} \setminus \mathcal{S}_{j'}} \frac{\exp(b_j) - \exp(b_{j'})}{\sum_{i' \in [k]} \exp(f_{i'}(x))} + O(\delta^r)$$

$$\leq \frac{k}{N} \sum_{x \in \mathcal{S}_j} \frac{\exp(b_j) - \exp(b_{j'})}{\sum_{i' \in [k]} \exp(f_{i'}(x))} + O(\delta^r)$$

$$\leq -\Omega(\mu \delta^r) \frac{k}{N} \sum_{x \in \mathcal{S}_j} \frac{\exp(b_j)}{\sum_{i' \in [k]} \exp(f_{i'}(x))} + O(\delta^r).$$

Similar as in Lemma 1, we can show that $\frac{\exp(b_j)}{\sum_{i' \in [k]} \exp(f_{i'}(x))} \geq \Omega(1)$ due to $W_{j,j} \leq O(\delta)$ and $b_j \geq \max_{i' \in [k]} b_{i'} - O(\delta^r)$. So, finally we have

$$\dot{b}_{j'} - \dot{b}_j \leq -\Omega(\mu \delta^r) + O(\delta^r) < 0,$$

where the last inequality chooses $\mu$ as a large enough constant. □

**Lemma 15** (Accuracy Monotonicity). *Given any positive constant $C_2$, there exists positive constant $C_1$ such that for all different $i, j \in [k]$, as long as $W_{i,i} \geq C_1$ and $f_i(x) - f_j(x) \leq C_2$ for any $x \in \mathcal{S}_i$, we have $\frac{\partial (f_i(x) - f_j(x))}{\partial t} > 0$.*

**Proof of Lemma 15.** Since $f_i(x) - f_j(x) \leq C_2$, we know $1 - u_i(x) \geq \Omega(1)$. This immediately implies $\min_{x' \in \mathcal{S}_i} (1 - u_i(x')) \geq \Omega(1)$ since $|u_i(x) - u_i(x')| \leq O(\delta)$. According to Lemma 13, we can bound $\frac{\partial (f_i(e_i) - f_j(e_i))}{\partial t}$ as follows,

$$\frac{\partial (f_i(x) - f_j(x))}{\partial t} \geq \Omega(W_{i,i}^{2r-2}) - O(1) > 0$$

where the second inequality holds because $W_{i,i} \geq C_1$ with $C_1$ a large enough constant. □

**Lemma 3** (Bias Gap Control II). *There exist small positive constants $C_1, C_2$ such that for any $j \in [k-1]$ and any $x \in \mathcal{S}_j$, if $1 - u_j(x) \leq C_1$, $W_{k,k} \leq O(\delta)$ and $b_j - b_k \geq -C_2$, we have $\dot{b}_j - \dot{b}_k < -\Omega(1)$.*

**Proof of Lemma 3.** Since $1 - u_j(x) \leq C_1$ for some $x \in \mathcal{S}_j$, we know $1 - u_j(x') \leq C_1 + O(\delta)$ for every $x' \in \mathcal{S}_j$. We can write down $\dot{b}_j - \dot{b}_k$ as follows,

$$\dot{b}_j - \dot{b}_k = \left(1 - \frac{k}{N} \sum_{x' \in \mathcal{S}} u_j(x')\right) - \left(1 - \frac{k}{N} \sum_{x' \in \mathcal{S}} u_k(x')\right)$$

$$= \frac{k}{N} \sum_{x' \in \mathcal{S}_j} (u_k(x') - u_j(x')) + \frac{k}{N} \sum_{x' \in \mathcal{S} \setminus \mathcal{S}_j} (u_k(x') - u_j(x')).$$

First, we upper bound $u_k(x') - u_j(x')$ for every $x' \in \mathcal{S}_j$ as follows,

$$u_k(x') - u_j(x') \leq 1 - u_j(x') - u_j(x') = -1 + 2\left(1 - u_j(x')\right) \leq 2C_1 - 1 + O(\delta).$$

Same as in the proof of Lemma 1, for each $x' \in \mathcal{S} \setminus \mathcal{S}_j$, we can bound $u_j(x'), u_k(x')$ as follows,

$$\frac{\exp\left(b_j\right)}{\sum_{i' \in [k]} \exp\left(f_{i'}(x')\right)} - O(\delta^r) \leq u_j(x') \leq \frac{\exp\left(b_j\right)}{\sum_{i' \in [k]} \exp\left(f_{i'}(x')\right)} + O(\delta^r),$$

$$\frac{\exp\left(b_k\right)}{\sum_{i' \in [k]} \exp\left(f_{i'}(x')\right)} - O(\delta^r) \leq u_k(x') \leq \frac{\exp\left(b_k\right)}{\sum_{i' \in [k]} \exp\left(f_{i'}(x')\right)} + O(\delta^r).$$

Therefore, we can upper bound $u_k(x') - u_j(x')$ as follows,

$$
\begin{aligned}
u_k(x') - u_j(x') &\leq \frac{\exp\left(b_k\right) - \exp\left(b_j\right)}{\sum_{i' \in [k]} \exp\left(f_{i'}(x')\right)} + O(\delta) \\
&= \frac{\exp\left(b_j\right)}{\sum_{i' \in [k]} \exp\left(f_{i'}(x')\right)} \cdot \left(\exp(b_k - b_j) - 1\right) + O(\delta^r) \\
&\leq O(C_2) + O(\delta^r),
\end{aligned}
$$

where the last inequality uses $b_k - b_j \leq C_2$.

Above all, we can upper bound $\dot{b}_j - \dot{b}_k$ as follows,

$$
\begin{aligned}
\dot{b}_j - \dot{b}_k &\leq -1 + 2C_1 + O(C_2) + O(\delta^r) \\
&< -\Omega(1),
\end{aligned}
$$

where the second inequality holds as long as $C_1, C_2, \delta$ are at most some small constants. $\qquad\square$

**Lemma 14** (Bias Gap Control III). *For any different $i, j \in [k]$, if $W_{i,i} \leq O(1), W_{j,j} \leq O(\delta)$ and $b_i - b_j \leq -O(1)$, we have*

$$\dot{b}_i - \dot{b}_j > 0.$$

**Proof of Lemma 14.** We can write down $\dot{b}_i - \dot{b}_j$ as follows,

$$
\begin{aligned}
\dot{b}_i - \dot{b}_j &= \left(1 - \frac{k}{N} \sum_{x \in \mathcal{S}} u_i(x)\right) - \left(1 - \frac{k}{N} \sum_{x \in \mathcal{S}} u_j(x)\right) \\
&= \frac{k}{N} \sum_{x \in \mathcal{S}} \left(u_j(x) - u_i(x)\right)
\end{aligned}
$$

Next, we lower bound $u_j(x) - u_i(x)$ for every $x \in \mathcal{S}$,

$$
\begin{aligned}
u_j(x) - u_i(x) &= \frac{\exp\left(\langle W_{j,:}, x\rangle^r + b_j\right) - \exp\left(\langle W_{i,:}, x\rangle^r + b_i\right)}{\sum_{i' \in [k]} f_{i'}(x)} \\
&\geq \frac{\exp\left(O(\delta^r) + b_j\right) - \exp\left(O(1) + b_i\right)}{\sum_{i' \in [k]} f_{i'}(x)}
\end{aligned}
$$

So as long as $b_j - b_i > O(1)$, we have $u_j(x) - u_i(x) > 0$ for all $x \in \mathcal{S}$, which then implies $\dot{b}_i - \dot{b}_j > 0$. $\qquad\square$

**Lemma 17.** *For every $j \in [k]$ and every $x \in \mathcal{S}$, we have*

$$\left|\left\langle \dot{W}_{j,:}, \xi_x \right\rangle\right| \cdot T \leq O\left(\sqrt{\log N}\sigma\delta\right)$$

**Proof of Lemma 17.** For each $j \in [k]$, we have

$$\dot{W}_{j,:} = \frac{k}{N} \left(\sum_{x' \in \mathcal{S}_j} \left(1 - u_j(x')\right) r \langle W_{j,:}, x'\rangle^{r-1} x' - \sum_{x' \in \mathcal{S} \setminus \mathcal{S}_j} u_j(x') r \langle W_{j,:}, x'\rangle^{r-1} x'\right)$$

and

$$
\left\langle \dot{W}_{j,:}, \bar{\xi}_x \right\rangle
$$

$$
= \frac{k}{N} \left( \sum_{x' \in \mathcal{S}_j} (1 - u_j(x')) \, r \, \langle W_{j,:}, x' \rangle^{r-1} \langle x', \bar{\xi}_x \rangle - \sum_{x' \in \mathcal{S} \setminus \mathcal{S}_j} u_j(x') r \, \langle W_{j,:}, x' \rangle^{r-1} \langle x', \bar{\xi}_x \rangle \right)
$$

We know that $|\langle x, \bar{\xi}_x \rangle| \leq O(\sigma + \sqrt{\log N}/\sqrt{d})$. For any $x' \neq x$, we have $|\langle x', \bar{\xi}_x \rangle| \leq O((\sigma\sqrt{\log N})/\sqrt{d} + \sqrt{\log N}/\sqrt{d}) \leq O(\sqrt{\log N}/\sqrt{d})$ as long as $\sigma \leq 1$.

According to Lemma 18, we know that for $x' \in \mathcal{S}_j$, we have $\left| (1 - u_j(x')) \langle W_{j,:}, x' \rangle^{r-1} \right| \leq O(1)$. For $x' \in \mathcal{S} \setminus \mathcal{S}_i$, we have $\left| u_i(x') \langle W_{j,:}, x' \rangle^{r-1} \right| \leq O(\delta^{r-1})$ since $|\langle W_{j,:}, x' \rangle| \leq O(\delta) + O(\sqrt{\log N}\delta\sigma) \leq O(\delta)$ assuming $\sigma \leq 1/\sqrt{\log N}$.

Therefore, we can bound $\left| \left\langle \dot{W}_{j,:}, \bar{\xi}_x \right\rangle \right|$ as follows,

$$
\left| \left\langle \dot{W}_{j,:}, \bar{\xi}_x \right\rangle \right| \leq O \left( \frac{\sigma}{N} + \frac{\sqrt{\log N}}{\sqrt{d}} \right)
$$

Since $T \leq O(\log(1/\delta)/\delta^{r-2})$, $N \geq \log(1/\delta)/\delta^{r-1}$ and $d \geq \log^2(1/\delta)/\delta^{2r-2}$, we know

$$
\left| \left\langle \dot{W}_{j,:}, \bar{\xi}_x \right\rangle \right| \cdot T \leq O(\sqrt{\log N}\delta).
$$

$\square$

**Lemma 18.** *For any $i \in [k]$ and $x \in \mathcal{S}_i$, if $(1 - u_i(x)) \langle W_{i,:}, x \rangle^{r-1} \geq \Theta(1)$, we have*

$$
\frac{d}{dt} \left( (1 - u_i(x)) \langle W_{i,:}, x \rangle^{r-1} \right) < 0.
$$

**Proof of Lemma 18.** We can write $1 - u_i(x)$ as $\frac{\sum_{j \in [k], j \neq i} \exp(f_j(x))}{\sum_{j \in [k], j \neq i} \exp(f_j(x)) + \exp(f_i(x))}$. Next, we prove that for any $j' \neq i$, we have

$$
\frac{d}{dt} \left( \frac{\exp(f_{j'}(x))}{\sum_{j \in [k], j \neq i} \exp(f_j(x)) + \exp(f_i(x))} \langle W_{i,:}, x \rangle^{r-1} \right) < 0.
$$

This derivative can be written the sum of two terms:

$$
\frac{d}{dt} \left( \frac{\exp(f_{j'}(x))}{\sum_{j \in [k], j \neq i} \exp(f_j(x)) + \exp(f_i(x))} \langle W_{i,:}, x \rangle^{r-1} \right)
$$

$$
= \frac{1}{\sum_{j \in [k], j \neq i} \exp(f_j(x) - f_{j'}(x)) + \exp(f_i(x) - f_{j'}(x))} \frac{d}{dt} \left( \langle W_{i,:}, x \rangle^{r-1} \right)
$$

$$
+ \frac{d}{dt} \left( \frac{1}{\sum_{j \in [k], j \neq i} \exp(f_j(x) - f_{j'}(x)) + \exp(f_i(x) - f_{j'}(x))} \right) \langle W_{i,:}, x \rangle^{r-1}.
$$

For the first term, we have

$$
\frac{1}{\sum_{j \in [k], j \neq i} \exp(f_j(x) - f_{j'}(x)) + \exp(f_i(x) - f_{j'}(x))} \frac{d}{dt} \left( \langle W_{i,:}, x \rangle^{r-1} \right)
$$

$$
= \frac{1}{\sum_{j \in [k], j \neq i} \exp(f_j(x) - f_{j'}(x)) + \exp(f_i(x) - f_{j'}(x))} (r-1) \langle W_{i,:}, x \rangle^{r-2} \left\langle \dot{W}_{i,:}, x \right\rangle
$$

$$
\leq \frac{1}{\exp(f_i(x) - f_{j'}(x))} (r-1) \langle W_{i,:}, x \rangle^{r-2} \left\langle \dot{W}_{i,:}, x \right\rangle.
$$

For the second term, we have

$$\frac{d}{dt}\left(\frac{1}{\sum_{j\in[k],j\neq i}\exp\left(f_j(x)-f_{j'}(x)\right)+\exp\left(f_i(x)-f_{j'}(x)\right)}\right)\langle W_{i,:},x\rangle^{r-1}$$

$$=-\frac{\sum_{j\in[k],j\neq i}\exp\left(f_j(x)-f_{j'}(x)\right)\left(\dot{f}_j(x)-\dot{f}_{j'}(x)\right)+\exp\left(f_i(x)-f_{j'}(x)\right)\left(\dot{f}_i(x)-\dot{f}_{j'}(x)\right)}{\left(\sum_{j\in[k],j\neq i}\exp\left(f_j(x)-f_{j'}(x)\right)+\exp\left(f_i(x)-f_{j'}(x)\right)\right)^2}\langle W_{i,:},x\rangle^{r-1}$$

$$\leq-\frac{1}{2}\cdot\frac{r\langle W_{i,:},x\rangle^{r-1}\left\langle\dot{W}_{i,:},x\right\rangle}{\exp\left(f_i(x)-f_{j'}(x)\right)}\langle W_{i,:},x\rangle^{r-1},$$

where the last inequality uses $f_i(x)-f_j(x)\geq\Omega(1)$, $\left|\dot{f}_j(x)-\dot{f}_{j'}(x)\right|\leq O(1)$ and $\dot{f}_i(x)-\dot{f}_{j'}(x)\geq r\langle W_{i,:},x\rangle^{r-1}\left\langle\dot{W}_{i,:},x\right\rangle-O(1)\geq\Omega(1)$.

Combining the bounds on both terms, as long as $\langle W_{i,:},x\rangle$ is larger than certain constant (which is guaranteed by $(1-u_i(x))\langle W_{i,:},x\rangle^{r-1}\geq\Theta(1)$), we know $\frac{d}{dt}\left((1-u_i(x))\langle W_{i,:},x\rangle^{r-1}\right)<0$. $\square$

**Lemma 16.** *For any $j\neq j'$, we have $T\dot{W}_{j,j'}\leq O(\delta)$. Furthermore, there exists absolute constant $\mu>0$ such that if $0<W_{j,j'}<\frac{\mu\delta}{\log^{\frac{1}{r-2}}(1/\delta)}$, we have $T\dot{W}_{j,j'}\geq-\frac{\mu\delta}{2\log^{\frac{1}{r-2}}(1/\delta)}$.*

**Proof of Lemma 16.** We can write down the derivative of $W_{j,j'}$ as follows,

$$\dot{W}_{j,j'}$$
$$=\frac{k}{N}\sum_{x\in\mathcal{S}_j}(1-u_j(x))r\langle W_{j,:},x\rangle^{r-1}\langle e_{j'},x\rangle$$
$$-\frac{k}{N}\sum_{x\in\mathcal{S}_{j'}}u_j(x)r\langle W_{j,:},x\rangle^{r-1}\langle e_{j'},x\rangle$$
$$-\frac{k}{N}\sum_{x\in\mathcal{S}\setminus(\mathcal{S}_j\cup\mathcal{S}_{j'})}u_j(x)r\langle W_{j,:},x\rangle^{r-1}\langle e_{j'},x\rangle$$
$$=\pm O\left(\frac{\sigma\sqrt{\log N}}{\sqrt{d}}\right)-O\left(\left(W_{j,j'}\pm O\left(\sqrt{\log N}\delta\sigma\right)\right)^{r-1}\right)\pm O\left(\delta^{r-1}\frac{\sigma\sqrt{\log N}}{\sqrt{d}}\right).$$

The bound on the first term relies on $(1-u_j(x))\langle W_{j,:},x\rangle^{r-1}\leq O(1)$ and $\langle e_{j'},x\rangle=\pm O\left(\frac{\sigma\sqrt{\log N}}{\sqrt{d}}\right)$ for $x\in\mathcal{S}_j$, where $(1-u_j(x))\langle W_{j,:},x\rangle^{r-1}\leq O(1)$ is guaranteed by Lemma 18. The bound on the second term uses $\langle W_{j,:},x\rangle=W_{j,j'}\pm O\left(\sqrt{\log N}\delta\sigma\right)$ and $\langle e_{j'},x\rangle=1\pm O\left(\frac{\sigma\sqrt{\log N}}{\sqrt{d}}\right)$ for $x\in\mathcal{S}_{j'}$. The bound on the third term uses $\langle W_{j,:},x\rangle=O(\delta)$ and $\langle e_{j'},x\rangle=\pm O\left(\frac{\sigma\sqrt{\log N}}{\sqrt{d}}\right)$ for $x\in\mathcal{S}\setminus(\mathcal{S}_j\cup\mathcal{S}_{j'})$.

To prove the upper bound of the derivative, we have

$$\dot{W}_{j,j'}\leq O(\frac{\sigma\sqrt{\log N}}{\sqrt{d}})$$

where we use $W_{j,j'}\pm O\left(\sqrt{\log N}\delta\sigma\right)\geq 0$. Since $T=O(\log(1/\delta)/\delta^{r-2})$, we have

$$T\dot{W}_{j,j'}\leq O(\delta),$$

as long as $d\geq O(\frac{\log N\log^2(1/\delta)}{\delta^{2r-2}})$.

We show that there exists absolute constant $\mu > 0$ such that if $0 < W_{j,j'} < \frac{\mu\delta}{\log^{\frac{1}{r-2}}(1/\delta)}$, we have

$T\dot{W}_{j,j'} \geq -\frac{\mu\delta}{2\log^{\frac{1}{r-2}}(1/\delta)}$, which holds as long as $\dot{W}_{j,j'} \geq -O\left(\frac{\mu\delta^{r-1}}{\log^{\frac{r-1}{r-2}}(1/\delta)}\right)$. We have

$$\dot{W}_{j,j'}$$
$$= \pm O(\frac{\sigma\sqrt{\log N}}{\sqrt{d}}) - O\left(\left(W_{j,j'} \pm O\left(\sqrt{\log N}\delta\sigma\right)\right)^{r-1}\right)$$
$$\geq -O(\frac{\sigma\sqrt{\log N}}{\sqrt{d}}) - O\left(\frac{\mu^{r-1}\delta^{r-1}}{\log^{\frac{r-1}{r-2}}(1/\delta)}\right)$$
$$\geq -O\left(\frac{\mu^{r-1}\delta^{r-1}}{\log^{\frac{r-1}{r-2}}(1/\delta)}\right)$$
$$\geq -O\left(\frac{\mu\delta^{r-1}}{\log^{\frac{r-1}{r-2}}(1/\delta)}\right).$$

The first inequality assumes $\sigma \leq O\left(\frac{\mu}{\sqrt{\log N}\log^{\frac{1}{r-2}}(1/\delta)}\right)$. The second inequality assumes $d \geq O(\frac{\log N \log^{\frac{2r-2}{r-2}}(1/\delta)}{\mu^{2r-2}\delta^{2r-2}})$. The third inequality chooses $\mu$ as a small enough constant. $\quad\square$

**Lemma 11.** *For any $j \in [k]$, we have*

$$\frac{\partial W_{j,j}^{(t)}}{\partial t} \geq -O\left(\frac{\delta^{r-1}\sqrt{\log N}\sigma}{\sqrt{d}}\right).$$

*If $\min_{x \in \mathcal{S}_j}(1 - u_j(x)) \geq \Omega(1)$, we further have*

$$\left(1 - O(\sqrt{\log N}\sigma)\right)\frac{k}{N}\sum_{x \in \mathcal{S}_j}(1 - u_j(x))rW_{j,j}^{r-1} \leq \frac{\partial W_{j,j}^{(t)}}{\partial t} \leq \left(1 + O(\sqrt{\log N}\sigma)\right)\frac{k}{N}\sum_{x \in \mathcal{S}_j}(1 - u_j(x))rW_{j,j}^{r-1}.$$

**Proof of Lemma 11.** We have

$$\dot{W}_{j,j} = \frac{k}{N}\sum_{x \in \mathcal{S}_j}(1 - u_j(x))r\langle W_{j,:}, x\rangle^{r-1}\langle x, e_j\rangle - \frac{k}{N}\sum_{x \in \mathcal{S}\setminus\mathcal{S}_j}u_j(x)r\langle W_{j,:}, x\rangle^{r-1}\langle x, e_j\rangle$$

$$= \frac{k}{N}\sum_{x \in \mathcal{S}_j}(1 - u_j(x))r\left(W_{j,j} \pm O\left(\sqrt{\log N}\delta\sigma\right)\right)^{r-1}\left(1 \pm O\left(\frac{\sqrt{\log N}\sigma}{\sqrt{d}}\right)\right)$$

$$\quad - \frac{k}{N}\sum_{x \in \mathcal{S}\setminus\mathcal{S}_j}u_j(x)r\left(O(\delta) \pm O\left(\sqrt{\log N}\delta\sigma\right)\right)^{r-1}\left(\pm O\left(\frac{\sqrt{\log N}\sigma}{\sqrt{d}}\right)\right)$$

$$= \left(1 \pm O(\sqrt{\log N}\sigma)\right)\frac{k}{N}\sum_{x \in \mathcal{S}_j}(1 - u_j(x))rW_{j,j}^{r-1} \pm O\left(\frac{\delta^{r-1}\sqrt{\log N}\sigma}{\sqrt{d}}\right),$$

where the second equality uses $|\langle W_{j,:}, \xi_x\rangle| \leq O(\sqrt{\log N}\delta\sigma), |\langle \xi_x, e_j\rangle| \leq O\left(\frac{\sqrt{\log N}\sigma}{\sqrt{d}}\right)$ and $W_{j,j'} \leq O(\delta)$ for $j \neq j'$.

Therefore, if $\min_{x \in \mathcal{S}_j}(1 - u_j(x)) \geq \Omega(1)$, we know

$$\left(1 - O(\sqrt{\log N}\sigma)\right)\frac{k}{N}\sum_{x \in \mathcal{S}_j}(1 - u_j(x))rW_{j,j}^{r-1} \leq \dot{W}_{j,j} \leq \left(1 + O(\sqrt{\log N}\sigma)\right)\frac{k}{N}\sum_{x \in \mathcal{S}_j}(1 - u_j(x))rW_{j,j}^{r-1}.$$

And we always have

$$\dot{W}_{j,j} \geq -O\left(\frac{\delta^{r-1}\sqrt{\log N}\sigma}{\sqrt{d}}\right).$$

$\quad\square$

**Lemma 13.** *For any $x \in \mathcal{S}_i$ and any $j \neq i$, if $1 - u_i(x) \geq \Omega(1)$, we have*

$$\frac{\partial}{\partial t}\left(f_i(x) - f_j(x)\right) \geq \Omega(W_{i,i}^{2r-2}) - O(1).$$

**Proof of Lemma 13.** Recall that $f_i(x) = \langle W_{i,:}, x \rangle^r + b_i$, so we have

$$\begin{aligned}
\dot{f}_i(x) =& r \langle W_{i,:}, x \rangle^{r-1} \left\langle \dot{W}_{i,:}, x \right\rangle + \dot{b}_i \\
=& r \left( W_{i,i} \pm \sqrt{\log N} \sigma \delta \right)^{r-1} \left( \dot{W}_{i,i} + \left\langle \dot{W}_{i,:}, \xi_x \right\rangle \right) + \dot{b}_i \\
\geq& \Omega(W_{i,i}^{2r-2}) - O(1),
\end{aligned}$$

where in the last inequality we uses $\dot{W}_{i,i} \geq \Omega(W_{i,i}^{r-1}) \geq \Omega(\delta^r)$ and $\left| \left\langle \dot{W}_{i,:}, \xi_x \right\rangle \right| \leq \frac{\sigma \sqrt{\log N} \delta^{r-1}}{\log(1/\delta)} \leq O(\delta^{r-1})$.

We also have

$$\begin{aligned}
\dot{f}_j(x) =& r \langle W_{j,:}, x \rangle^{r-1} \left\langle \dot{W}_{j,:}, x \right\rangle + \dot{b}_j \\
=& r \left( W_{j,i} + \sqrt{\log N} \sigma \delta \right)^{r-1} \left( \dot{W}_{j,i} \pm \left\langle \dot{W}_{j,:}, \xi_x \right\rangle \right) + \dot{b}_j \\
\leq& O(1),
\end{aligned}$$

where we uses $|W_{j,i}| \leq O(\delta), \left| \dot{W}_{j,i} \right| \leq O(\delta^{r-1})$ and $\left| \left\langle \dot{W}_{j,:}, \xi_x \right\rangle \right| \leq O(\delta^{r-1})$.

Therefore, we have

$$\frac{dt}{d}\left(f_i(x) - f_j(x)\right) \geq \Omega(W_{i,i}^{2r-2}) - O(1).$$

$\square$

## D   ADDITIONAL EXPERIMENTS

In this section, we describe the detailed setting of our experiments and also include additional experiment results.

**MNIST & Fashion-MNIST.**    Unless specified otherwise, we use a depth-10 and width-1024 fully-connected ReLU neural network (FCN10) for MNIST and Fashion-MNIST. We use Kaiming initialization for the weights and set all bias terms as zero. We use a small initialization by scaling the weights of each layer by $(0.001)^{1/h}$ so the output is scaled by $0.001$, where $h$ is the network depth. We train the network using SGD with learning rate $0.01$ and momentum $0.9$ for 100 epochs.

**CIFAR-10 & CIFAR-100**   We use VGG-16 (without batch normalization) for CIFAR-10 and CIFAR-100. We use Kaiming initialization for the weights and set all bias terms as zero. We run SGD with momentum $0.9$ and weight decay $1e-4$ for 100 epochs. For the learning rate, we start from $0.01$ and reduce it by a factor of $0.1$ at the 60-th epoch and 90-th epoch.

We linearly interpolate using 50 evenly spaced points between the network at initialization and the network at the end of training. We evaluate error and loss on the train set. For each setting, we repeat the experiments three times from different random seeds and plot the mean and deviation.

Note in Figure 1, to contrast the convex curve and plateau curve, we have used FCN4 with standard initialization on MNIST, and VGG-16 with 0.001 initialization on CIFAR-10.

Our code is based on the implementation from Lucas et al. (2021). Each trial of our experiment can be finished on an Nvidia Tesla P100 within one hour.

## D.1 ALL BIAS V.S. LAST BIAS V.S. NO BIAS

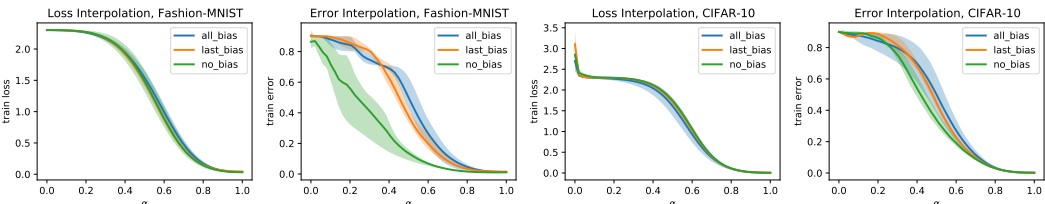

Figure 9: Comparison between networks with all bias, last bias and no bias on Fashion-MNIST and CIFAR-10.

Figure 9 shows that on both Fashion-MNIST and CIFAR-10, having bias on the last layer or on all layers can create longer plateau in error curve, while does not significantly affect the loss curve.

## D.2 NORMAL INTERPOLATION V.S. HOMOGENEOUS INTERPOLATION.

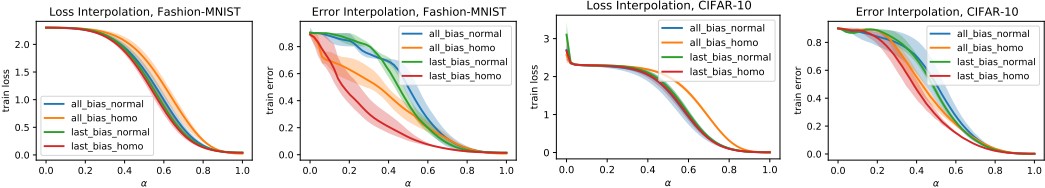

Figure 10: Comparison between networks with normal interpolation and homogeneous interpolation on bias on Fashion-MNIST and CIFAR-10.

Figure 10 shows that on both Fashion-MNIST and CIFAR-10, applying homogeneous interpolation on biases can significantly reduce the plateau on error interpolation curve.

## D.3 DIFFERENT INITIALIZATIONS

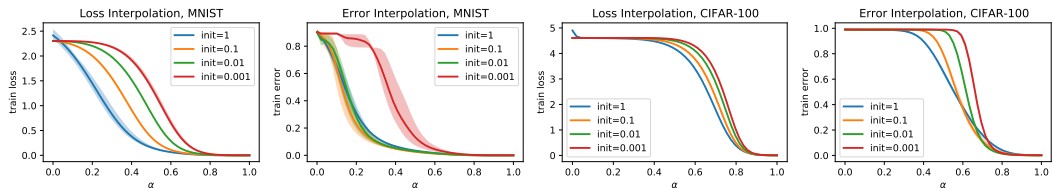

Figure 11: Comparison between networks with different initialization scales on MNIST and CIFAR-100 with last bias.

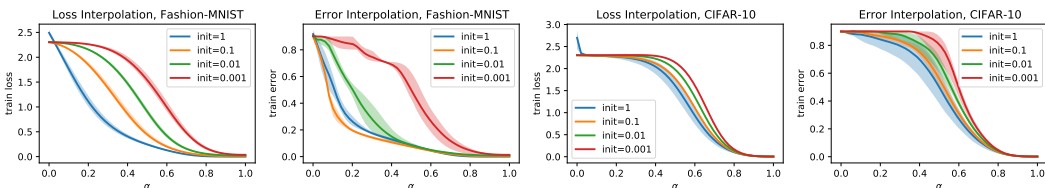

Figure 12: Comparison between networks with different initialization scales on Fashion-MNIST and CIFAR-10 with all bias.

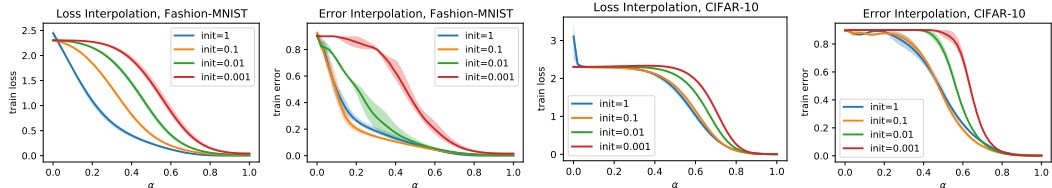

Figure 13: Comparison between networks with different initialization scales on Fashion-MNIST and CIFAR-10 with last bias.

Smaller initialization creates longer plateau in both error and loss curves. See Figure 11 for MNIST, CIFAR-100 with last bias; see Figure 12 for Fashion-MNIST, CIFAR-10 with all bias; see Figure 13 for Fashion-MNIST, CIFAR-10 with last bias.

## D.4 DIFFERENT DEPTHS

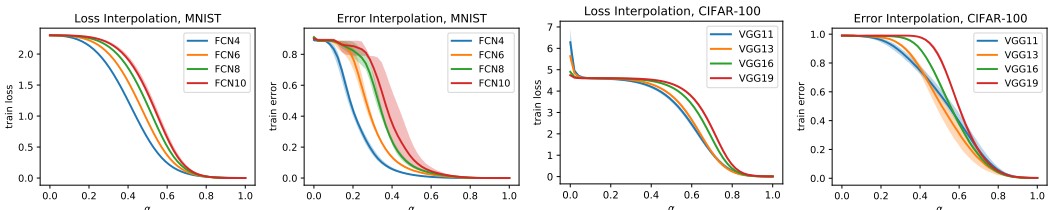

Figure 14: Comparison between networks with different depth on MNIST and CIFAR-100 with last bias.

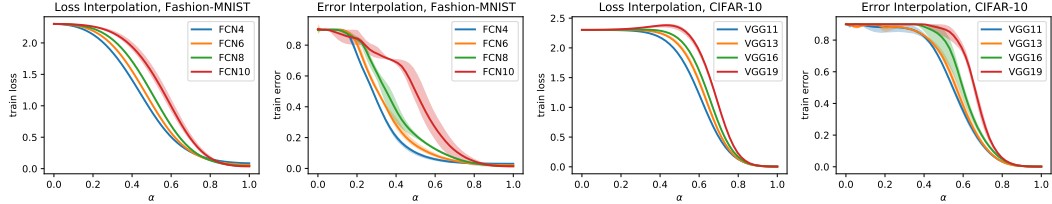

Figure 15: Comparison between networks with different depth on Fashion-MNIST and CIFAR-10 with all bias. We use 0.001 initialization scale for VGG-16 on CIFAR-10.

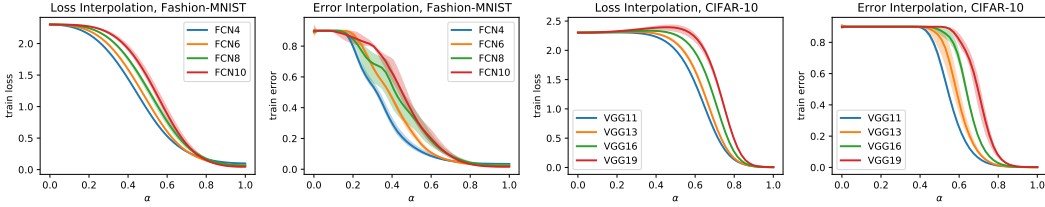

Figure 16: Comparison between networks with different depth on Fashion-MNIST and CIFAR-10 with last bias. We use 0.001 initialization scale for VGG-16 on CIFAR-10.

Deeper networks create longer plateau in both error and loss curves. See Figure 14 for MNIST, CIFAR-100 with last bias; see Figure 15 for Fashion-MNIST, CIFAR-10 with all bias; see Figure 16 for Fashion-MNIST, CIFAR-10 with last bias.

### D.5 Bias dynamics

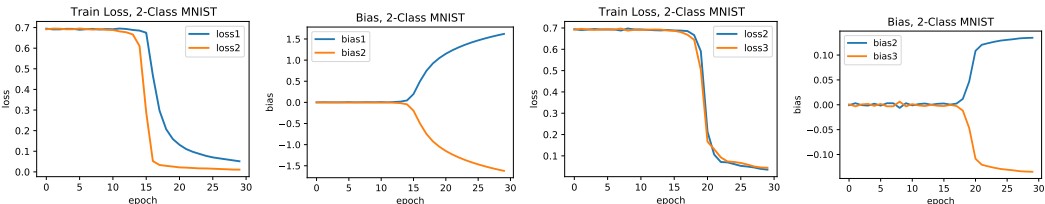

Figure 17: Train loss for each class and bias term dynamics on MNIST$\{1, 2\}$ and MNIST$\{2, 3\}$.

In Figure 17, we give two more examples on two-class MNIST in which the later learned class has larger bias.

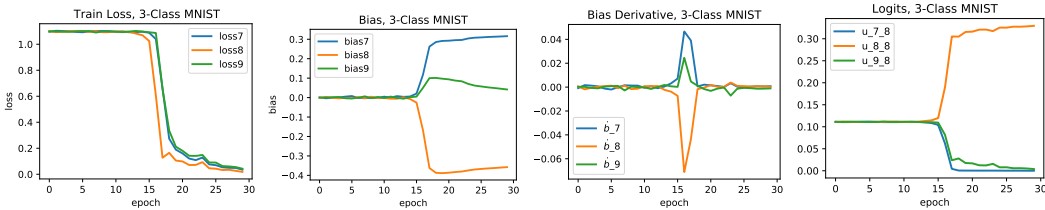

Figure 18: Train loss for each class and bias term dynamics on MNIST$\{7, 8, 9\}$.

In Figure 18, although class 9 is learned last, class 7 gets the largest bias after training. Let $S$ be the set of all samples for number 7,8,9 and let $S_7, S_9, S_9$ be the set of samples for each class. For convenience, we use $u_{i,j}$ to denote $\frac{1}{|S|} \sum_{x \in S_j} u_i(x)$, where $u_i(x)$ is the softmax output for class $i$ under input $x$. Then, we can write down the derivative on three bias terms:

$$
\dot{b}_7 = \frac{1}{3} - u_{7,7} - u_{7,8} - u_{7,9}
$$

$$
\dot{b}_8 = \frac{1}{3} - u_{8,7} - u_{8,8} - u_{8,9}
$$

$$
\dot{b}_9 = \frac{1}{3} - u_{9,7} - u_{9,8} - u_{9,9}.
$$

According to the per-class loss, we know that $\sum_{x \in S_7} -\log(u_7(x)) < \sum_{x \in S_9} -\log(u_9(x))$, which intuitively implies that $\sum_{x \in S_7} u_7(x) > \sum_{x \in S_9} u_9(x)$ that is $u_{7,7} > u_{9,9}$. This tends to drive $b_7$ smaller than $b_9$. However, because $u_{9,8} > u_{7,8}$, we actually have $\dot{b}_9 < \dot{b}_7$. So eventually $b_9$ becomes smaller than $b_7$. Intuitively, class 9 is more correlated with class 8, so $u_{9,8} > u_{7,8}$.

