# OpenReview forum: "Plateau in Monotonic Linear Interpolation --- A "Biased" View of Loss Landscape for Deep Networks"
_ICLR.cc/2023/Conference — ICLR 2023 poster_

### Official Review · Reviewer_snjJ · 2022-10-20

**Confidence:** 3
**Correctness:** 3
**Technical Novelty And Significance:** 3
**Empirical Novelty And Significance:** 3
**Recommendation:** 6

**Clarity, Quality, Novelty And Reproducibility:**

Overall, the paper is clear, although the flow of the narrative can be improved (e.g. placing of figures in relation to the text). The theoretical work is thorough, and the experiments are well put together. In my opinion, this work is of high quality.  The observation (relating plateaus to biases) is novel, however, it remains to be seen whether the discovery of this relationship will be impactful. The correlation between network depth, initialisation range and the plateaus seems to support the standard interpretation of MLI that the authors set out to debunk.

**Strength And Weaknesses:**

Strengths:
* The paper challenges the naive use of a crude estimation to make far-reaching conclusions. I strongly resonate with the message: do not over-interpret, aim to fully understand first.
* The results are supported by both theory and practice. The experiments are realistic and do not over-simplify the application domain.

Weaknesses:
* The overall presentation flow is a bit incoherent, with numerous past and future references to sections and figures. I found myself furiously scrolling up and down in an attempt to follow the plot.
* The authors “attribute the plateau to simple reasons as the bias terms, the network initialization scale and the network depth”. Specifically for the last two, it is shown that smaller initialisation and larger depth lead to bigger plateaus. But is it really that unfair to correlate these with problem difficulty? Intuitively, weights initialised in a smaller region would yield weaker gradients in a deep network, which may slow down the learning = make the problem harder. Similarly, deeper networks experience vanishing gradients, which once again increases the difficulty of learning. Stating these as “simple reasons” sounds like an oversimplification.
* The paper contains a few small grammatical and formatting mistakes, please refer to the following non-exhaustive list of suggested corrections:

		Page 2: “Under this model we can show” - the sentence ends abruptly with no punctuation, consider adding “…that:” or otherwise rephrasing.
		Page 4: “achieve loss approximately” -> achieve a loss of approximately…
		Page 5: “detailed proof in Section C” - do you mean Appendix C?
		Appendix A: “We give two examples that illustrates” -> that illustrate

**Summary Of The Paper:**

The paper shows that monotonic linear interpolations (MLIs), commonly used to study the loss landscapes of neural networks (NNs), may be misleading if used to estimate problem difficulty. Specifically, it is shown (both theoretically and empirically) that the plateaus on the MLI plots, attributed to problem “hardness”, are in fact related to the bias training dynamics. The authors argue that the flatness of the MLI curve can be easily manipulated by tweaking hyper parameters such as initialisation range and network depth. The authors also make an interesting observation that the class that happens to be learned last by the NN ends up with the strongest (largest) bias.

**Summary Of The Review:**

Overall, I think this is a good paper that may yield interesting discussions in the community. Developing theory of NN learning is crucial for the field, and this work adds to the overall understanding, even if in a humble way.

---

> ### Author Response · Authors · 2022-11-09
> **Response**
>
> Thank you so much for your detailed comments and positive review! We are glad to answer your questions here.
>
> *“Small initialization and deep network can cause vanishing gradient, which increases the difficulty of training”* \
> We agree that if one uses a very small initialization on a very deep network, the gradient can vanish and the training becomes difficult. However, in our experiments, for the fully-connected neural network, we only increase the depth from 4 to 10 and re-scale the standard initialization by $(0.001)^{1/10}\approx 0.50$; for VGG, we only increase the depth from 11 to 19 and re-scale the standard initialization by $(0.001)^{1/16}\approx 0.65$. These modest changes do not seem to affect training difficulty in practice, but they significantly increase the plateau length in the loss and error interpolations.
>
> Thanks again for your comments and suggestions! We have also fixed other formatting/grammatical issues.

---

### Official Review · Reviewer_jXBe · 2022-10-23

**Confidence:** 3
**Correctness:** 3
**Technical Novelty And Significance:** 4
**Empirical Novelty And Significance:** 3
**Recommendation:** 8

**Clarity, Quality, Novelty And Reproducibility:**

Clarity: The paper is in general well-written with clear notation. The problem is well-explained and motivated and the story line is very clear.

Quality: The theoretical experiments consider a rather simple setting. However, it is still quite challenging to derive meaningful theoretical results in those simplified setting. The theoretical predictions from those experiments seem to transfer to more realistic settings, as the carefully conducted experiments show.

Novelty: To the best of my knowledge, the results in this paper are novel and go beyond existing literature.

Reproducibility: I could not have a close look into the proofs, but the proof strategy which was suggested in the main text seems to be  very reasonable to me.


**Strength And Weaknesses:**

Strenghts: The problem in this paper is well-motivated and the paper is very nice to read. The explanations are convincing both in theory and practice. I think that this is a paper which should be of interest to the ICLR community.

Weaknesses: The toy model is a bit simplistic but I think that this is OK since our understanding of neural network training is still very limited.

Typos and minor issues:
1. p. 4: "$V_{\max}$for all" (blank space missing)
2. p. 5, third line: "where (the) weight vector"
3. p. 5: The work of Ge et al., 2021 is about tensor decomposition. Consider explaining how it connects to the setting in this paper and, in particular, to the chosen activation function.
4. p. 5, statement of Theorem 2: "Suppose the neural network, dataset and optimization procedure are as defined in Section 2." Are these things really all defined in Section 2? I cannot find them there. Please consider clarifying.
5. p. 5, Theorem 2, statements 1 and 2: By using the notation $\Omega (1)$, do you mean that there is a small constant $c>0$ such that these statements hold? Otherwise please clarify.
6. p. 7: In the second statement of Theorem 3, consider replacing "monotonically decreasing" with "strictly monotonically decreasing"

Questions:
1. p. 4, Theorem 1: It seems to me that Theorem 1 makes only Assumption 1 regarding the (fully) trained neural network.
Is that really true? Do you not also need an assumption which says that the loss function is small at the end? Or am I missing something? Otherwise please consider adding the missing assumptions to the theorem.
2. p. 4: "Empirically, $\frac{\Delta_{\min}}{2 V^r_{\max}}$... does not change much when depth increases." Do you mean that you empirically notice that increases depth increases $\Delta_{\min}$?
3. p. 5, beginning of page: Why do you need $r\ge 3$? Why would $r=2$ not suffice?
4. Suppose all biases would be equal after the network has been fully trained. Why can't you then prove that there is a plateau? There would still be the different scaling, i.e. $\theta$ vs. $\theta^r$. Consider adding this explanation to the main text.

**Summary Of The Paper:**

This paper considers the problem of monotonic linear interpolation (MLI) in deep neural networks. That is, one connects the initialization of the neural network with the fully trained neural network through a line. Then one considers how the loss function behaves when one is interpolation between the weights at initialization and the weights at the fully trained neural networks. What has been observed is that in modern architectures, the loss stays first for some time at a plateau and then suddenly decreases as one is moving from the initialization to the fully trained neural network.

This paper argues that the reason for the reason for this phenomenon is that the biases in the last layer scale linearly with the interpolation parameter $\theta$, whereas all the other parameter scale with $\theta^r$, where $r$ denotes the number of layers. Importantly, they claim that there is one bias in the last layer which is much larger than all the others.

The paper makes the following contribution which corroborates these claims.
1. They prove that if there is bias which is much larger than the others, then the phenomenon described above appears.
2. They prove that in a simple toy setting there is indeed one bias which is larger than all the others (and they can subsequently use the results to strengthen the results in point 1 for this simple toy model.)
3. They provide numerical experiments which corroborate 1. and 2.

**Summary Of The Review:**

I think that this is a good paper, which has good theoretical and numerical experiments and which gives insight into an interesting question regarding the behaviour. of neural network training.
For these reasons, I think that this is a good paper which should be accepted.

---

> ### Author Response · Authors · 2022-11-09
> **Response**
>
> Thank you so much for your detailed comments and positive review! We are glad to answer your questions here.
>
> *“Why does Theorem 1 only require the bias gap?”* \
> In Theorem 1, we just proved the plateau in loss and error interpolations, which only requires the bias gap assumption. One might need stronger assumptions on the final loss and accuracy to prove the monotonicity of loss and error interpolations, for example in Theorem 3.
>
> *“$\frac{\Delta_{\min}}{2V_{\max}^r}$ does not change much when depth increases”* \
> In Theorem 1, we use $\frac{\Delta_{\min}}{2V_{\max}^r}$ to lower bound the ratio between the bias gap and the weights signal. In our empirical experiments, when the network depth changes, the ratio between the bias gap and the weights signal roughly remains the same.
>
> *“Why do you need $r\geq 3$?”* \
> In the training dynamics analysis of the r-homogeneous-weight network, we need different classes to be learned at very different times. That is, when $W_{i,i}$ rises to a constant, we need $W_{j,j}$ for $j>i$ to remain small. We prove this by showing that $W_{i,i}$’s behave as tensor power iterations when $W_{i,i}$’s are small, which requires $r\geq 3$ (see Lemma 12). When $r = 2$ the weights behave like matrix power iterations and the gap does not increase fast enough.
>
> *“What about all biases being equal in the end?”* \
> When all biases are equal, the bias terms all cancel out in the soft-max computation, which is equivalent to zero biases. In this case, we still have the plateau in the loss interpolation for a deep network under small initialization since the weights signal remains small. However, there is no plateau in the error interpolation since the interpolated network output is basically a scaled version of the network
> output at the minimum and has the same label predictions.
>
> *“What’s the connection with tensor decomposition in Ge et al., 2021?”* \
> The activation function of our r-homogeneous-weight network is a polynomial of degree r, which can be represented by tensors of order r. In our work, we assume the features $v_i$’s to be orthogonal for the convenience of training dynamics analysis. For similar reasons, Ge et al., 2021 assume the ground truth tensor is orthogonally decomposable.
>
> *“The reference of ‘Section 2’ in Theorem 2?”* \
> Sorry, this is a typo, it should be Section 4 instead.
>
> *“$\Omega(1)$ in Theorem 2”* \
> When using $\Omega(1)$, we mean these statements hold for a quantity that’s at least a positive constant.
>
> Thanks again for your comments and suggestions! We have revised our paper accordingly and also fixed other minor issues.

---

### Official Review · Reviewer_K8b8 · 2022-10-24

**Confidence:** 3
**Correctness:** 4
**Technical Novelty And Significance:** 3
**Empirical Novelty And Significance:** 3
**Recommendation:** 8

**Clarity, Quality, Novelty And Reproducibility:**

The paper is written clearly, the proofs are presented understandably, despite their complexity. The experiments are described well and code is provided, so that results are reproducible. The theoretical results and their empirical support are novel.

**Strength And Weaknesses:**

Strength:
- strong theoretical contribution
- empirical support for theoretical results
- insightful conclusion that the shape of the loss curve along the linear interpolation is not indicative of the success of training or the simplicity of the optimization problem

Weaknesses:
- the results on training dynamics hold only for a two-layer FCNN with specific activation
- Thm .1 does not apply to the r-homogeneous-weight network used in the following section, since it requires a neural network with at least 3 layers

**Summary Of The Paper:**

The paper analyses the shape of the loss curve along the linear interpolation from initialization to minimum for neural networks. In particular, in analyses under which circumstances the loss curve is monotonic and when it exhibits a plateau. The paper shows theoretically that for fully-connected neural networks the existence of the plateau can be linked to the gap in last-layer biases and the network's initialization, independent of the difficulty of the optimization problem. Moreover, the paper analyzes the training dynamics wrt. this bias gap. These theoretical results are supported by empirical evidence on MNIST and CIFAR100.

**Summary Of The Review:**

The paper presents novel theoretical results on the loss curve of the linear interpolation between initialization and minimum. These results indicate that the hardness of the learning problem plays little role in the existence of a plateau. Instead, the existence of a plateau can be linked to the bias terms, which in term can be linked to initialization and network depth. This is a valuable insight. The examples in the appendix that underpin this insight are great.

The theoretical analysis is limited to specific types of networks (fully-connected feed-forward networks with at least three layers and only an output bias for Sec 3 and two-layer fully-connected feed-forward networks with a specific polynomial activation function for Sec 4), but the empirical results are performed on standard DNNs (including VGG16). It would be great if the results in Sec. 3 and 4 could be linked, though, because now the assumptions on the network are mutually exclusive.

Nonetheless, the paper is insightful and relevant to the community, technically sound, and well-written.

Detailed comments:
- In Thm. 2: network as defined in Section 2 -> should be section 4
- r is used both as number of layers in Sec 3 and as parameter of the activation function in Sec 4, this is a it confusing

---

> ### Author Response · Authors · 2022-11-09
> **Response**
>
> Thank you so much for your detailed comments and positive review! We are glad to answer your questions here.
>
> *“Theorem 1 does not apply to the r-homogeneous-weight network”* \
> Although Theorem 1 does not directly apply to the r-homogeneous-weight network, we have analyzed the plateau for the r-homogeneous-weight network in Theorem 3. The results in Theorem 3 are actually stronger with tighter bounds for the plateau and also a monotonicity analysis.
>
> *“Assumptions in Sections 3&4 are mutually exclusive”* \
> We consider a fully-connected ReLU neural network in Section 3. And in Section 4, we consider a simpler model (the r-homogeneous-weight network) for the convenience of the training dynamics analysis. These two models might seem very different, but the r-homogeneous-weight network essentially simulates a depth-r network with only last-layer biases in the sense that the weights are r-homogenous and the bias is 1-homogenous. More precisely, for both models, scaling all the weights by a factor of $c$ and scaling all the biases by a factor of $c^r$ scales the output of the entire network by a factor of $c^r$.  The plateau analyses in Section 3 and Section 4 are also very similar.
>
> *“$r$ is used as the number of layers in Section 3 but as the parameter of the activation in Section 4”* \
> As we explained above, we use the r-homogeneous-weight network to simulate a depth-r network with only last-layer biases in the sense that the weights are r-homogenous and the bias is 1-homogenous. Therefore, we use the same symbol for the number of layers for fully-connected ReLU/linear neural networks and the activation parameter for the r-homogeneous-weight network. We clarified this in the revised version of the paper.
>
> Thanks again for your comments and suggestions! We have revised our paper accordingly and also fixed other minor issues.

---

### Official Review · Reviewer_KBHX · 2022-10-25

**Confidence:** 3
**Correctness:** 4
**Technical Novelty And Significance:** 4
**Empirical Novelty And Significance:** 3
**Recommendation:** 6

**Clarity, Quality, Novelty And Reproducibility:**

As previously mentioned, the paper provides new insights into the MLI property and I believe it contains enough novelty.

The paper is clearly written. I did not fully check the proof but it looks technically sound to me.

**Strength And Weaknesses:**

Overall I enjoy reading this paper. The paper proposes a novel view of the MLI property, which is related to the last-layer biases. The observation is not previously explained, and could be a good supplementary of the existing theory of MLI. I personally like the implicit idea presented by the authors that a seemingly common optimization property is actually "not common", but dependent on specific factors instead. This might explain why it is so difficult to obtain a common but useful theory for neural nets.

I have some questions regarding the results in the paper and would like to learn the authors' opinions.

1. The paper's setting considers a classification problem with a cross-entropy loss. Do authors expect similar phenomenon happens in other problems such as regression problems, or with a different type of loss function?

2. It seems that by Theorem 1, we can only predict the length of the plateau after we train the network and obtain the final bias gap. Is there a way of knowing the length of the plateau before training (if the initialization is known)?

3. In the experiments, will changing the initialization affect the sequence of the class learned during training? For example, assume that in one experiment, class 1 is learned first and then class 2 is learned. By theory, the bias term in class 1 is smaller than class 2. Is it possible that class 2 turns to be learned first when we change a different initialization? In this case, the bias term in class 2 is smaller than class 1, and thus the two experiments result in very different outputs (although they may have similar loss or even similar training dynamics).

**Summary Of The Paper:**

This paper explores how plateau occur in the loss curves of neural networks. Specifically, it shows that a significantly different last-layer bias may result in a long plateau in the loss and error curves. It also analyzes how gradient dynamics generates such bias gap during training with a simplified classification model. Numerical experiments support the proposed theory.

**Summary Of The Review:**

In summary, I think the paper is in the right direction and presents new ideas. I recommend accepting this paper.

---

> ### Author Response · Authors · 2022-11-09
> **Response**
>
> Thank you very much for your valuable suggestions and positive review! We are glad to answer your questions here.
>
> *"The paper considers a classification problem with cross-entropy loss. What about regression problems?"* \
> We focus on classification problems with cross-entropy loss because this is the setting that most empirical works studied. We aim to explain the plateau in both loss and error interpolations (and the result for error interpolation is the more interesting one), while error/accuracy is usually only defined on a classification problem. That being said, we do believe some of our results can be extended to regression problems. In particular, on a deep network with a small initialization, the loss interpolation for regression problems should have a long plateau since the weights signal remains small.
>
> *“Is there a way to know the length of the plateau before training?”* \
> Our theory cannot predict the size of the bias gap, and therefore cannot precisely characterize the plateau length before training the network. But Theorem 1 does reveal interesting relations between the plateau length and several factors. In particular, the plateau for loss and error interpolations becomes longer under a deeper network or a smaller initialization; the plateau for error interpolation relies on the bias gap. These theoretical predictions were also verified in our experiments.
>
> *“Will changing the initialization affect the sequence of the classes learned during training?”* \
> Yes, this is entirely possible. The ordering of classes being learned depends on the initial correlations between the weights and class features. The larger the initial correlation with one class, the earlier this class gets learned. When all the classes have features with the same magnitude ($v_i$’s have the same norm as in our case), the ordering of the initial correlations and the ordering of classes being learned is uniformly random. On the other hand, if one class has a stronger feature (some $v_i$ has a larger norm than the other $v_j$’s), this class will be more likely to be learned first.
>
> Thanks again for your great questions! We hope our explanation has answered your questions.

---

### Decision · Program_Chairs · 2023-01-20

**Decision:**

Accept: poster

**Justification For Why Not Higher Score:**

The phenomenon found is not significant enough to be a spotlight paper.

**Justification For Why Not Lower Score:**

This paper provides new understanding about MLI and more generally, the optimization landscape of neural networks.

**Metareview: Summary, Strengths And Weaknesses:**

This paper studies the monotonic linear interpolation (MLI), which characterizes a proerty of the loss between a random initialization and the point GD converges to. While MLI was studied in a few works before, the work shows a new result  that the plateau of MLI is not necessiary related to the hardness of optimization, but related to the last-layer bais term theoretically and experimentally. This paper provides new understanding about MLI and more generally, the optimization landscape of neural networks. It is clearly written.  I recommend acceptance.

**Note From Pc:**

if the above contains the word "oral" or "spotlight" please see: "oral" presentation means -> notable-top-5% and "spotlight" means -> notable-top-25%. As stated in our emails, we are disassociating presentation type from AC recommendations